# A Bayesian approach to dynamic homology of morphological characters and the ancestral phenotype of jawed vertebrates

**Benedict King\*, Martin Rücklin**

Naturalis Biodiversity Center, Leiden, Netherlands

**Abstract** Phylogenetic analysis of morphological data proceeds from a fixed set of primary homology statements, the character-by-taxon matrix. However, there are cases where multiple conflicting homology statements can be justified from comparative anatomy. The upper jaw bones of placoderms have traditionally been considered homologous to the palatal vomer-dermopalatine series of osteichthyans. The discovery of 'maxillate' placoderms led to the alternative hypothesis that 'core' placoderm jaw bones are premaxillae and maxillae lacking external (facial) laminae. We introduce a BEAST2 package for simultaneous inference of homology and phylogeny, and find strong evidence for the latter hypothesis. Phenetic analysis of reconstructed ancestors suggests that maxillate placoderms are the most plesiomorphic known gnathostomes, and the shared cranial architecture of arthrodire placoderms, maxillate placoderms and osteichthyans is inherited. We suggest that the gnathostome ancestor possessed maxillae and premaxillae with facial and palatal laminae, and that these bones underwent divergent evolutionary trajectories in placoderms and osteichthyans.

**\*For correspondence:**
benking315@gmail.com

**Competing interests:** The authors declare that no competing interests exist.

## Introduction

The concept of homology underpins the cladistic analysis of morphological data. Testing of homology is usually considered a two-step process (*Patterson, 1982a*; *Pinna, 1991*). First, provisional statements of homology are made (primary homology), which are hypotheses based on comparative anatomy. Primary homologues are then subjected to cladistic analysis, and those that correspond to synapomorphies are then considered 'secondary homologues'; this term corresponds to the vernacular use of the term homology (similarity due to common ancestry). The starting point for a cladistic analysis, the character-by-taxon matrix, is a set of primary homology statements. Primary homology statements are based upon 'homology criteria' (*Patterson, 1988*; *Rutishauser and Moline, 2005*). The first and most important criterion for primary homology is similarity: structures should correspond in position and structural details (developmental similarity is part of this criterion). Second is the test of conjunction: if two structures are found together on a single animal, they cannot be homologous (*Patterson, 1988*).

Placoderms are stem gnathostomes, and the evolution and morphology of their jaws is thus of particular interest. The upper jaw bones of placoderms present a major unresolved example of a homology problem. Arthrodiran placoderms possess two upper gnathal plates in their jaws, termed the anterior and posterior supragnathals (*Figure 1A*). These have traditionally been considered primary homologues of the vomers and dermopalatines of osteichthyans (*Stensiö, 1963a*; *Stensiö, 1969*), which are palatal bones sitting on the roof of the mouth, inside the maxilla and premaxilla (*Figure 1C*). This proposed homology of placoderm supragnathals and osteichthyan palatal bones is based on positional criteria.

**Figure 1.** Upper jaw bones in arthrodire placoderms, maxillate placoderms and osteichthyans, showing alternative homology assignments for the arthrodiran supragnathals. (A–B) Arthrodire in palatal view, showing anterior and posterior supragnathals. Based on *Hu et al., 2017*. (C) Osteichthyan *Eusthenopteron* in palatal view, based on *Jarvik, 1980*. (D) Maxillate placoderm *Qilinyu* in palatal view, based on *Zhu et al., 2016*. Blue coloration indicates the premaxilla-maxilla series, red coloration indicates the vomer-dermopalatine series. The alternative coloration of arthrodire supragnathals in A and B represents the alternative homology statements for these bones (homology states 0 and 1 respectively).

The discovery of maxillate placoderms reignited debates about the homology of placoderm and osteichthyan skull bones (*Zhu et al., 2013*; *Zhu et al., 2016*), and a new hypothesis regarding the homology of arthrodiran gnathal plates was proposed (*Zhu et al., 2016*; *Zhu et al., 2019*). Maxillate placoderms have premaxillae and maxillae with both palatal and facial laminae (*Figure 1D*). The palatal laminae articulate with the ventral surface of the braincase, and therefore correspond in position to arthrodiran supragnathals. The facial laminae are continuous with the external dermal bones of the skull, and are equivalent in position to osteichthyan premaxillae/maxillae. *Zhu et al., 2016* therefore proposed the homology of arthrodiran supragnathals with the premaxilla and maxilla of

osteichthyans. This negates a putative homology with the osteichthyan vomer-dermopalatine series, which would otherwise fail the test of conjunction (placoderm supragnathals cannot be homologous to both the premaxilla-maxilla and vomer-dermoplatine series). Nevertheless, the traditional hypothesis for the homology of arthrodiran supragnathals continues to be discussed in the literature (*Hu et al., 2017*). There are therefore two opposing possibilities for the primary homology of arthrodiran gnathal bones.

A number of approaches have been proposed to distinguish between conflicting hypotheses of primary homology. *Jardine, 1969* provided a method that selected between alternative homologies of rhipidistian skull roof bones without reference to phylogeny, based on the criterion of preservation of spatial relationship. *Lee, 1998* used parsimony to distinguish between conflicting conjectures of homology on a fixed tree topology. The latter was essentially the approach taken by *Zhu et al., 2016* to support their hypothesis regarding placoderm supragnathal bones. However, choices regarding primary homology statements necessarily restrict the search for secondary homologues: phylogenetic analyses can only find the optimal tree given the input character matrix. Indeed, it has been suggested that the two-step approach to homology entails a degree of circularity (*Rieppel, 1996*), although this is likely to only be an issue when a phylogeny is weakly supported. A solution to this issue is the simultaneous inference of primary and secondary homology, termed *dynamic homology.*

Dynamic homology of molecular sequence data in a parsimony framework has been implemented in the software POY (*Wheeler et al., 2006*; *Varón et al., 2010*). Models for dynamic homology of molecular data have also been developed (*Lunter et al., 2005*; *Redelings and Suchard, 2005*; *Wheeler, 2006*) and implemented within the phylogenetic software Bali-Phy (*Suchard and Redelings, 2006*) and POY 5.0 (*Wheeler et al., 2015*). *Agolin and D'Haese, 2009*, used the parsimony implementation in POY to analyze morphological data (specifically the setae of collembolans). However, morphological characters, with their hierarchical dependence relationships and arbitrary sequence within a data matrix, are often not amenable to models used to align molecular data. *Ramírez, 2007* presented a parsimony approach to dynamic homology, using the empirical example of sclerites on the male copulatory organs of anyphaenid spiders. In this method, multiple matrices with alternative alignments of morphological characters were analysed, and the phylogenetic tree and homology combination with the shortest tree length was selected.

Dynamic homology methods for morphological data have thus far been rarely explored, and are restricted to parsimony-based approaches. However, a Bayesian approach would confer a number of advantages. Alternative homology statements could be considered as 'nuisance parameters', such that phylogenetic trees could be estimated while accounting for uncertainty in primary homology statements. Conversely, if discovering homology is the aim, the tree topology could be considered the 'nuisance parameter'. Bayesian tip-dated analysis of morphological data allows comparative analysis (such as biogeography or ancestral state reconstruction) to occur simultaneously with tree search (e.g. *Lee et al., 2018*). Comparative analyses could therefore be performed while accounting for uncertainty in both tree topology and primary homology statements.

Here, we present an approach to dynamic homology within a Bayesian tip-dating framework, which we use to test the alternative conjectures of placoderm jaw bone homologies. The homology relations of placoderm jaw bones have implications for our understanding of character evolution in early vertebrates. In particular, homology of placoderm supragnathal bones with the marginal jaw bones of osteichthyans suggests a deep (early) origin for these bones. *Zhu et al., 2016* proposed their hypothesis within the framework of placoderm paraphyly (*Brazeau, 2009*; *Davis et al., 2012*; *Zhu et al., 2013*), but an alternative hypothesis of placoderm monophyly (excluding maxillate placoderms) is supported by an essentially equivalent amount of morphological data, and is strongly supported under Bayesian tip-dated methods (*King et al., 2017*). The implications of the hypothesis of *Zhu et al., 2016* within the framework of placoderm monophyly have not been discussed. We therefore simultaneously estimated a credible set of phenotypes for the (apomorphy-defined) gnathostome common ancestor to explore character evolution in early gnathostomes while accounting for phylogenetic uncertainty, divergence date uncertainty, and alternative placoderm jaw bone homologies.

## Dynamic homology

We implemented a method for dynamic homology of morphological characters within the open source BEAST2 software package *homology* (https://github.com/king-ben/homology; **King, 2021**; copy archived at swh:1:rev:6e6dbd77443b0d963640b3cb603c4310b5a4b47e). The method takes as inputs alternative character coding alignments, here called *homology alignments*, which are alternative character codings corresponding to alternative homology hypotheses for morphological features (for example placoderm jaw bones). Homology alignments can be included alongside fixed alignments (*Figure 2*), such that only a subset of characters has dynamic homology. During a BEAST2 MCMC run, the homology alignment used to calculate the posterior is determined by a homology state parameter, which is changed by an operator (*Figure 2*). The MCMC will spend more time in the homology state corresponding to the homology alignment that returns the highest tree likelihood.

The *homology* package contains two java classes corresponding to CalculationNodes (which calculate a part of the posterior based on inputs). These are *HomologyTreeLikelihood* and *Homology-Multiplexer* (*Figure 3*). The *HomologyTreeLikelihood* class is an extension of the core BEAST2 *TreeLikelihood* class, and differs in associating a particular homology alignment with a homology state. The *HomologyMultiplexer* takes as input two or more HomologyTreeLikelihoods and a *homology parameter*, the latter is an integer parameter with states (0, 1,...,N) corresponding to N homology states (one for each HomologyTreeLikelihood). During an MCMC run, the homology-multiplexer returns the value of the homology tree likelihood corresponding to the current state of the homology parameter. Due to the possibility of correlated tree- and homology-space, the package also contains two updated tree operators which simultaneously change the tree topology and homology state: *HomologySAWilsonBalding* and *HomologySAExchange*.

## Results

Homoplasy-partitioned Bayesian tip-dated analysis (with dynamic homology of placoderm upper jaw bones) of the gnathostome fossil dataset results in the majority-rule consensus tree shown in *Figure 4*. Core placoderms (placoderms excluding maxillate forms) are monophyletic (posterior probability, pp = 1.0). The maxillate placoderms *Entelognathus* and *Qilinyu* are resolved as the sister group to core placoderms, but with weak support (pp = 0.70). *Janusiscus* is resolved as a stem osteichthyan, sister to *Dialipina*, but support for this grouping is again weak (pp = 0.57).

We find strong support for homology state 1 (pp = 0.984), corresponding to the hypothesis that placoderm supragnathal bones are homologous to premaxillae and maxillae (*Zhu et al., 2016*). The mean log likelihood for homology alignment 0 is −85.099, and for homology alignment 1 –79.883. The MCMC chain therefore rarely accepts proposals for homology state 0 (*Figure 5*).

Principal coordinates (PCO) analysis of gnathostome fossils reveals chondrichthyans (including acanthodians), osteichthyans and core placoderms form three discrete and well-separated groups (*Figure 6A*), concordant with the results of *Davis et al., 2012*. *Janusiscus* is an outlier, lying equidistant from the three groups, whereas maxillate placoderms plot close to core placoderms.

We used ancestral sequence logging in BEAST2 to reconstruct the phenotype of the gnathostome ancestor in each sample from the posterior. A sample of 90 of these reconstructed ancestors included in the PCO mostly plot close to placoderms, with a small number plotting in outlier positions closer to *Janusiscus*. A second PCO using only placoderms (maxillate and core) and the reconstructed ancestors is shown in *Figure 6B*, with the point cloud of reconstructed ancestors converted to a 2D density plot. *Entelognathus* plots close to the center of the ancestral area, while *Qilinyu*, arthrodires, petalichthyids and acanthothoracids are equidistant. Antiarchs and ptyctodontids plot the furthest from the reconstructed ancestors. However, it should be noted that the two principal axes account for less than 10% of the total variance.

Plotting the raw distance measures shows that maxillate placoderms are the most similar taxa to the reconstructed ancestors (*Figure 6C*). The individual taxon with the lowest distance to the reconstructed ancestor (in each sample from the posterior, n = 1801) was a maxillate placoderm for 95% of the reconstructed ancestors (*Figure 6D*). This suggests that of the known gnathostome fossils, the maxillate placoderms (in particular *Entelognathus*) are the least divergent known descendants of the gnathostome common ancestor.

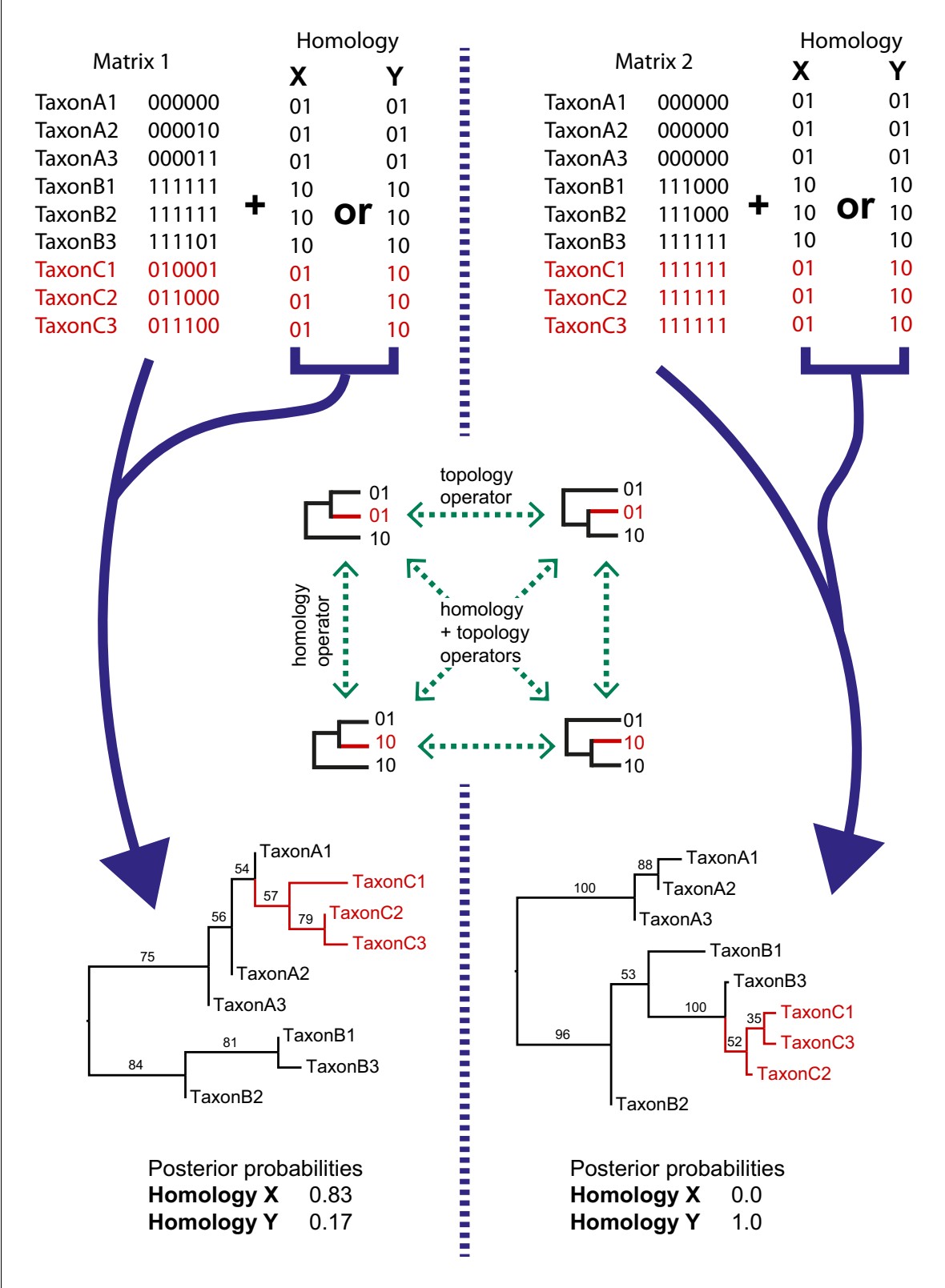

**Figure 2.** Simple examples of dynamic homology applied to matrices with six characters with fixed homology and two with estimated homology. Taxa C1–3 have alternative homologies (homology X and Y). For matrix1, there is moderate support for group C to fall within group A, leading to a higher posterior probability for homology X than homology Y. In matrix 2, there is strong support for taxon group C to fall within group B, leading in turn to strong support for homology Y.

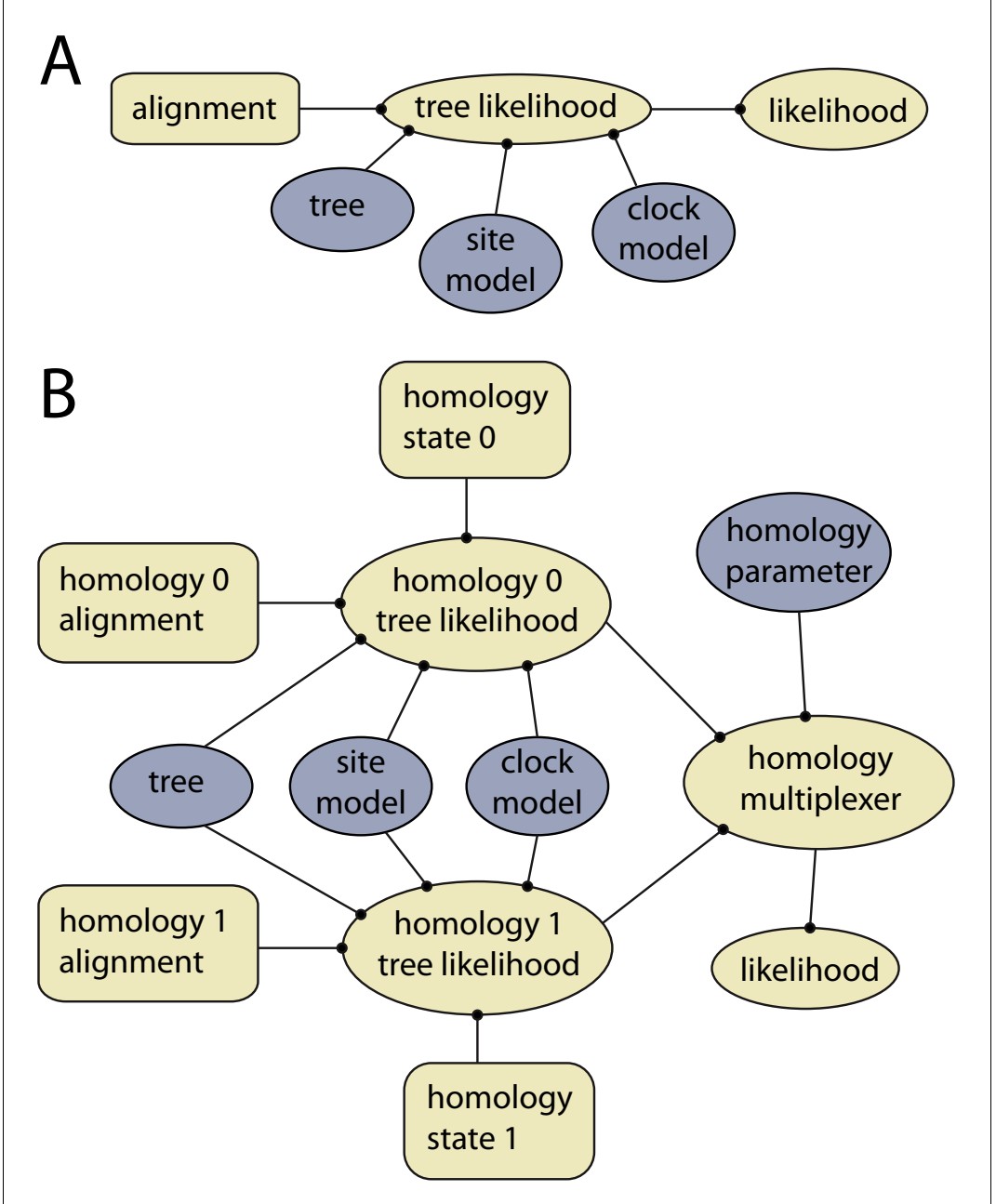

**Figure 3.** Example of models with and without dynamic homology of morphological characters. (**A**) Model diagram of a fixed homology partition. Tree likelihood takes as input a data alignment, tree, site model and clock model, and the calculated tree likelihood is passed to the likelihood, where it is combined with other partitions. Rectangles indicate fixed inputs, whereas ovals are model components that change during the MCMC. Blue shaded components are changed by operators, either directly (tree) or indirectly (site model, clock model). (**B**) Model diagram for a partition with dynamic homology with two homology states. The homology-multiplexer passes either the value of homology tree likelihood 0 or homology tree likelihood 1 to the likelihood, depending on the current value of the homology parameter.

The reconstructed ancestors also allow us to calculate the posterior probability of particular character states at the gnathostome node (i.e. the proportion of reconstructed ancestors with a particular character state). *Table 1* displays a number of characters of interest, including characters of the upper jaw bones and characters possessed by some core placoderms, argued to be retained plesiomorphies under the hypothesis of placoderm paraphyly (*Brazeau, 2009*; *Dupret et al., 2014*).

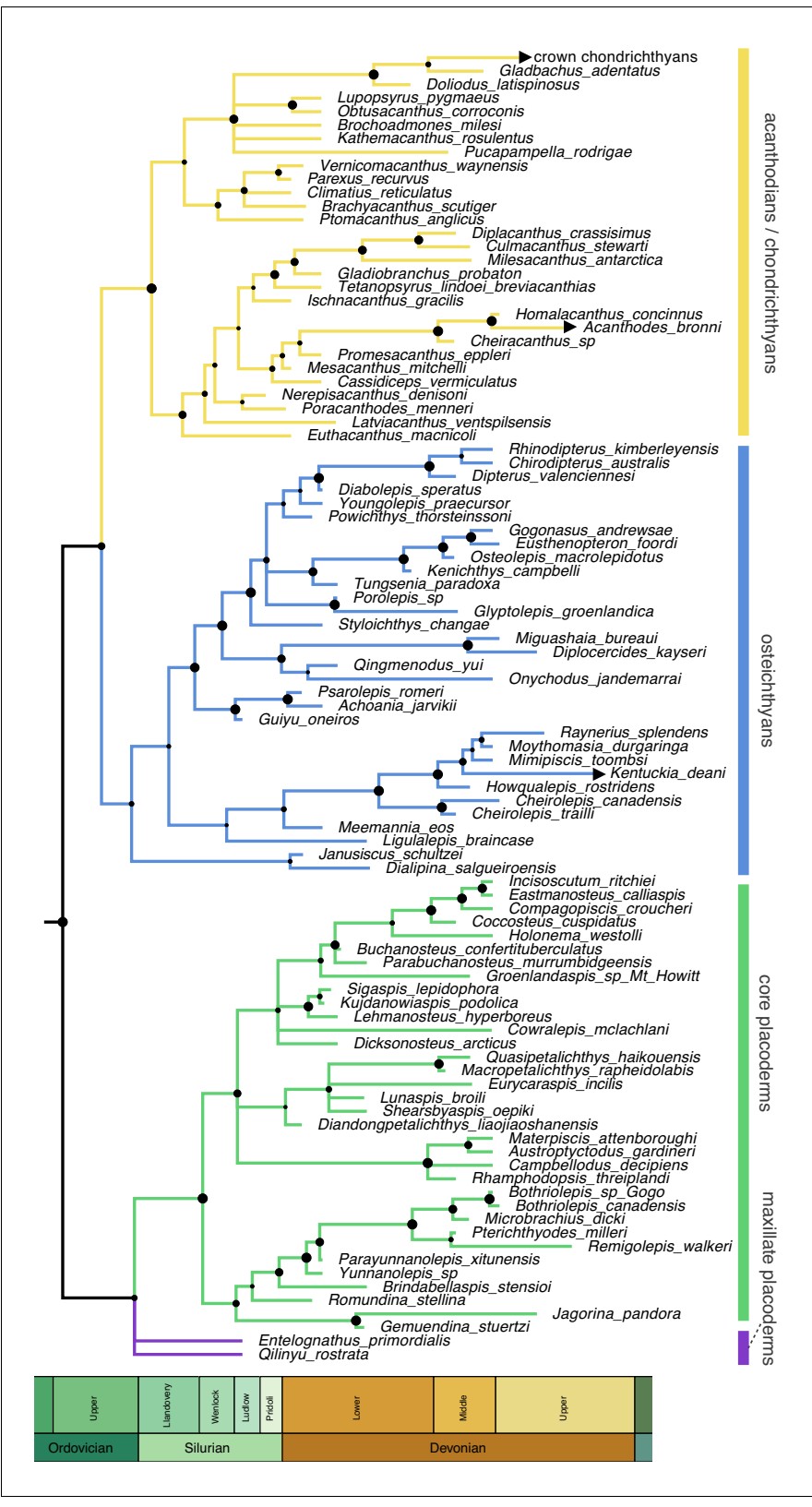

**Figure 4.** Time-scaled 50% majority-rule consensus tree from tip-dated homoplasy-partitioned analysis of gnathostome fossils, with dynamic homology of upper jaw bones in placoderms. Node circles indicate posterior probabilities. Branches with arrowheads (crown chondrichthyans, *Acanthodes*, *Kentuckia*) indicate tip age(s) are younger than the range displayed in the figure.

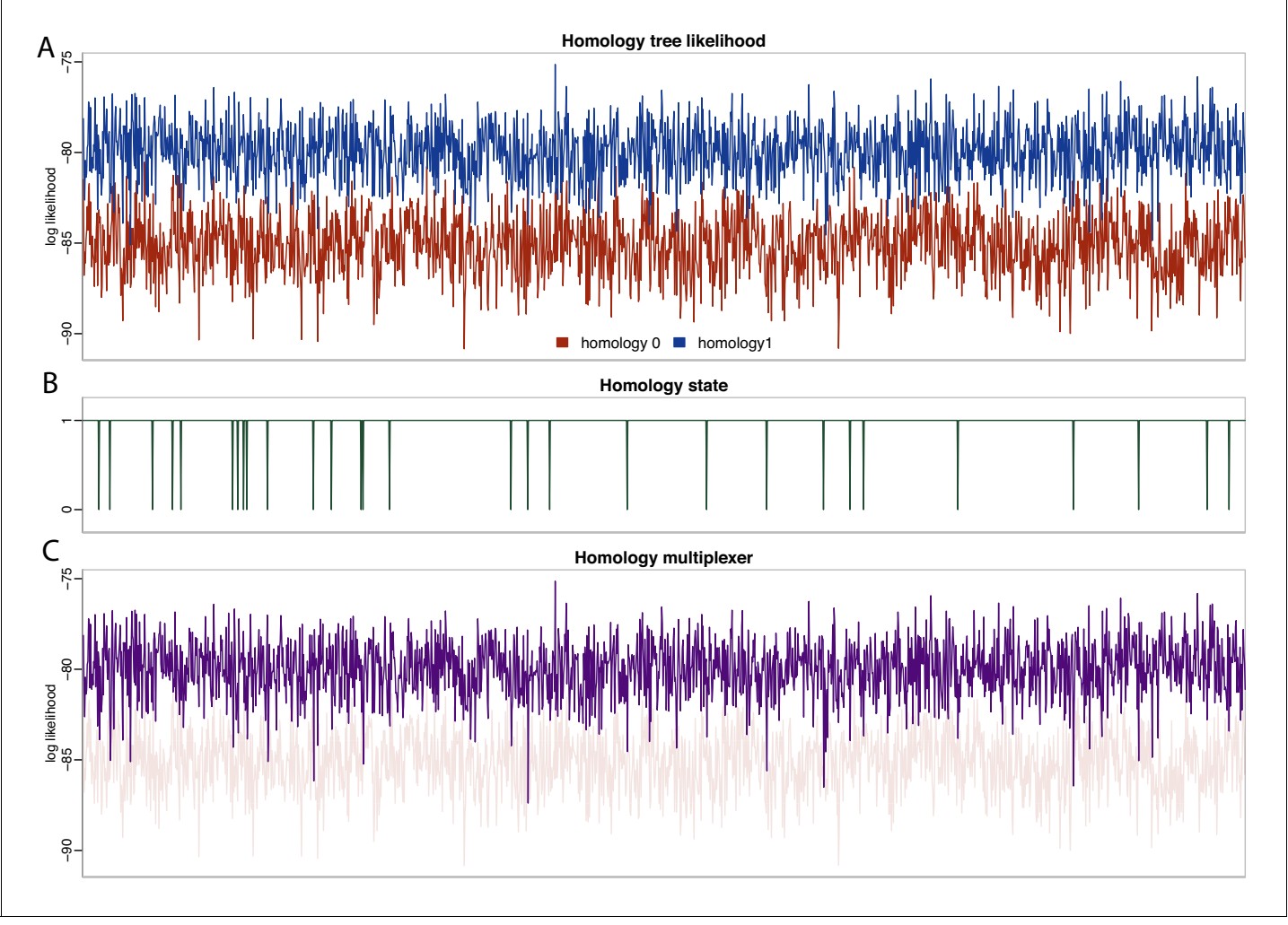

**Figure 5.** Likelihood and parameter traces during BEAST2 MCMC, with dynamic homology of placoderm jaw bones. (**A**) Tree likelihoods using homology alignment 0 (placoderm supragnathals are vomers/dermopalatines) are lower than those for homology alignment 1 (placoderm supragnathals are premaxillae/maxillae). (**B**) The MCMC only rarely samples homology state 0. (**C**) The homology-multiplexer therefore largely returns the tree likelihood of homology alignment 1 (homology tree likelihood 0 is replotted with transparency for reference).

Results for all characters are available in the supplementary information (*Table 1*; *Source data 1*). Our results suggest that the gnathostome ancestor had a premaxilla and maxilla with both palatal and facial laminae, no vomer-dermopalatine series, anterior/ventral nasal capsules and lateral orbits not surrounded by neurocranium. Putative core placoderm synapomorphies (claspers, optic fissure) are reconstructed as absent at the gnathostome node with moderate support (*Table 1*). This uncertainty is likely due to the high proportion of missing data for these characters. Critically, it is unknown whether or not maxillate placoderms possessed these putative core placoderm synapomorphies.

## Discussion

We find strong support for the hypothesis of *Zhu et al., 2016*, that placoderm supragnathal bones are homologous to the maxilla and premaxilla of osteichthyans and maxillate placoderms (*Figure 5*). However, we present a distinct scenario regarding the trajectory of upper jaw bone evolution (*Figure 7*). *Zhu et al., 2016* proposed that the plesiomorphic states of the maxillae and premaxillae were as palatal bones, exemplified by the arthrodiran condition. Facial laminae were then gained in the common ancestor of maxillate placoderms and crown gnathostomes, and palatal laminae were

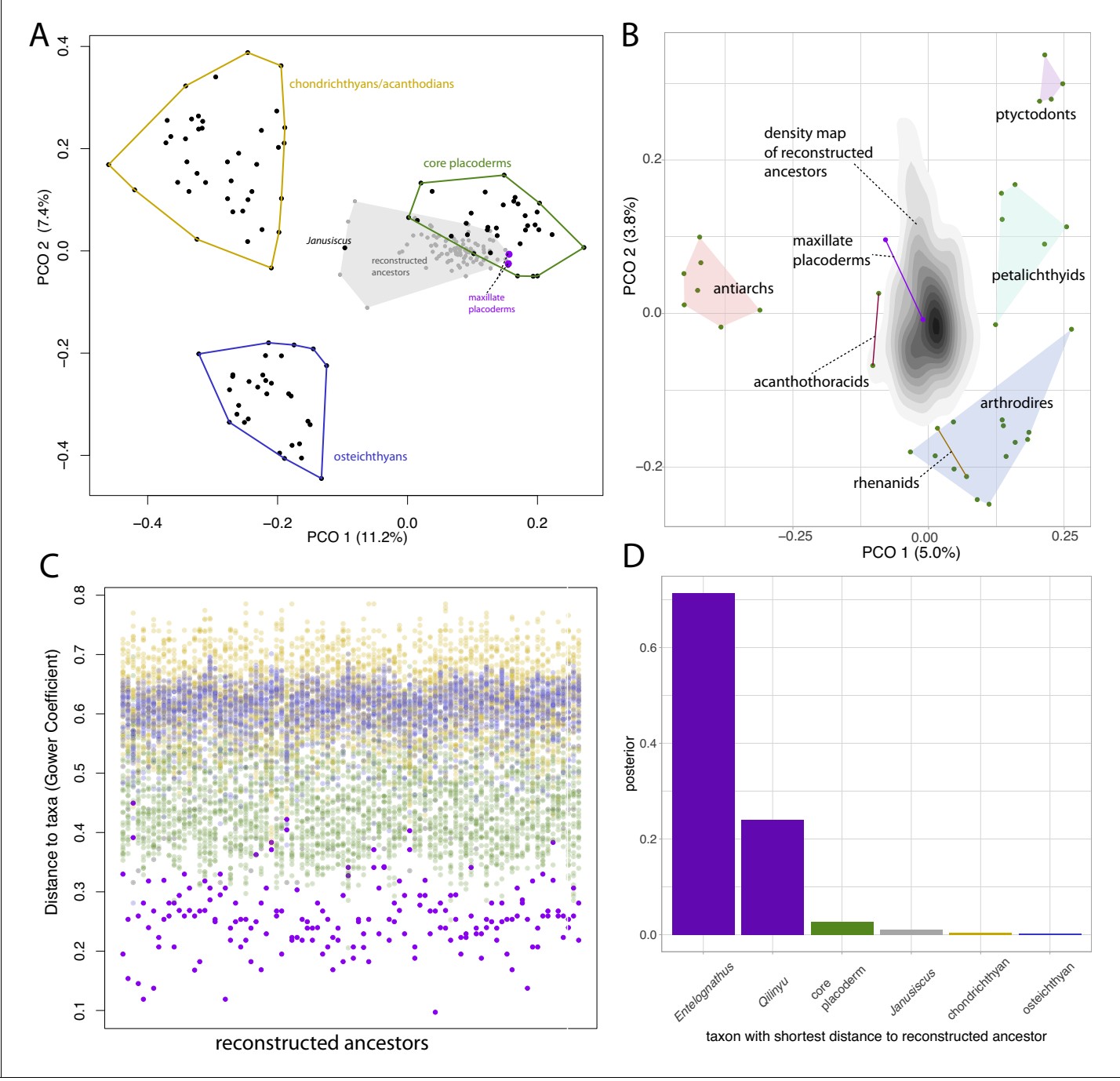

**Figure 6.** Distance plots suggest placoderms, and particularly maxillate placoderms are the gnathostomes least divergent from the gnathostome ancestor. (**A**) PCO plot of all gnathostome taxa in the matrix (black points) and a sample of 90 reconstructed ancestors (gray points and shaded convex hull). (**B**) PCO plot of placoderm taxa and reconstructed ancestors, with the latter point cloud converted to a density plot. (**C**) Each column represents a reconstructed ancestor (n = 90), with gnathostome fossils plotted with y-axis coordinates corresponding to distance from each reconstructed ancestor. (**D**) Frequency plot of the taxon with the shortest distance to the reconstructed ancestor across the whole posterior sample (n = 1801). Core placoderm, osteichthyan and chondrichthyan (including acanthodian) taxa are combined into single bins.

lost in osteichthyans. We instead propose that the common ancestor of (apomorphy-defined) gnathostomes possessed maxillae and premaxillae with both facial and palatal laminae. Facial laminae were subsequently lost in core placoderms and palatal laminae were lost in osteichthyans. The stem osteichthyans *Lophosteus* and *Andreolepis* show a possibly intermediate condition, in

**Table 1.** Character states reconstructed at the common ancestor of apomorphy-defined gnathostomes.

| Character | Reconstructed ancestral state | Posterior probability |
|---|---|---|
| Premaxilla | Present | 1.0 |
| Maxilla | Present | 0.96 |
| Facial laminae | Present | 0.96 |
| Palatal laminae | Present | 0.93 |
| Vomer | Absent | 0.93 |
| Dermopalatine | Absent | 0.95 |
| Nasal capsules | Anterior/ventral | 0.94 |
| Orbit dorsal, surrounded by neurocranium | Absent | 0.96 |
| Claspers | Absent | 0.79 |
| Optic fissure | Absent | 0.78 |

which the marginal jaw bones have internal (oral or palatal) laminae that are more strongly developed compared to other osteichthyans (*Botella et al., 2007*; *Cunningham et al., 2012*; *Chen et al., 2016*; *Chen et al., 2020*).

In concordance with *Zhu et al., 2016*, we find strong support for a lack of the vomer-dermopalatine series in the gnathostome ancestor. Our scenario suggests that arthrodires, for which morphological data of the jaws is best known (*Hu et al., 2017*), exhibit a specialized condition. Independent evidence for this hypothesis comes from recently described acanthothoracids (*Vaškaninová et al., 2020*), which exhibit marginal dentitions and jaw bones quite unlike those of arthrodires. In addition, the inner dental arcade of the stem osteichthyan *Lophosteus* consists of many 'tooth cushions' bearing no resemblance to arthrodire gnathal plates (*Chen et al., 2017*).

The divergent trajectories of the premaxilla and maxilla in osteichthyans and core placoderms may be associated with alternative ecological roles among their earliest members. Osteichthyans

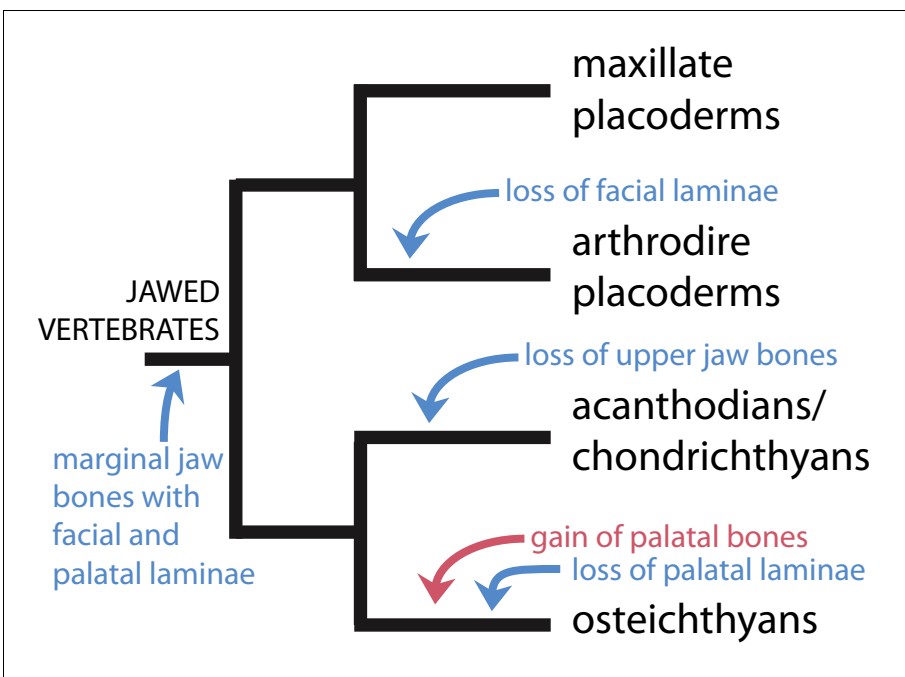

**Figure 7.** Scenario for the evolution of upper jaw bones in gnathostomes (jawed vertebrates). Red arrow indicates change to the palatal (vomer-dermopalatine) series of dermal jaw bones, blue arrows indicate changes to the marginal (premaxilla-maxilla) series.

from the Silurian Kuanti formation include the large *Megamastax* (*Choo et al., 2015*). The maxillate placoderms from the same formation however are clearly not apex predators, lacking large teeth on their jaw bones and in the case of *Entelognathus*, possess immovable eyes (*Zhu et al., 2013*). The loss of facial laminae in core placoderms may be associated with increased focus on crushing invertebrate prey, and may be analogous to the loss of the maxilla and specialization of the vomers in lungfishes. Conversely, the predatory osteichthyans emphasized the external tooth row and thus facial laminae.

Homology of the arthrodiran supragnathals with the premaxillae and maxillae of maxillate placoderms is consistent with observations from comparative anatomy (*Zhu et al., 2016*; *Zhu et al., 2019*). The snouts of maxillate placoderms differ from those of arthrodires mainly in the degree of dermal bone cover and are very similar in terms of their gross morphology. An early arthrodiran snout, such as that of *Kujdanowiaspis* (*Dupret, 2010*) differs from the maxillate placoderm condition by absence of facial laminae and a relatively small internasal plate compared to the large anterior premedian plate of *Entelognathus* (*Zhu et al., 2013*). *Zhu et al., 2019* suggested that the arthrodiran condition results from the inward shift of the upper jaw bones. However, the downturned, ventrally directed, snouts of maxillate placoderms means that reduction of the facial laminae and premedian plate are the only transformations required to leave the upper jaw bones separated from the dermal skull roof and in a palatal position, as in arthrodires.

The results of our phenetic analysis of reconstructed ancestors suggest maxillate-placoderm-like conditions in the last common ancestor of (apomorphy-defined) gnathostomes. Due to the nested position of acanthothoracids and antiarchs within a monophyletic core placoderms, we find strong support for anterior-ventral nasal capsules and lateral eyes in the gnathostome ancestor (*Table 1*). Under this hypothesis, the dorsal nasal capsules of antiarch, acanthothoracid and rhenanid placoderms are convergent with those of the jawless osteostracans and galeaspids, rather than representing shared plesiomophies (*King et al., 2017*). Conversely, the shared cranial architecture of arthrodires, maxillate placoderms and osteichthyans (*Dupret et al., 2014*), represent shared plesiomorphies (*Table 1*; *King et al., 2017*). Within agnathan fishes, the braincase proportions of the jawless heterostracans, which probably possess paired anterior nasal capsules (*Halstead, 1973*; *Janvier, 1996*), may represent the plesiomorphic gnathostome condition more closely than osteostracans or galeaspids.

Although our phenetic analysis suggests that maxillate placoderms are the gnathostomes morphologically closest to the ancestral condition, we are not suggesting that they are directly ancestral. The distance from each reconstructed ancestor is usually in the range 0.2–0.3, suggesting that even maxillate placoderms are highly derived from the gnathostome common ancestor. This result is not surprising given that our analysis suggests gnathostomes diverged during the Ordovician (*Figure 4*). Tentative support for this divergence might be found in the enigmatic fossils of *Skiichthys* (*Smith and Sansom, 1997*) and Mongolepidae (suggested to be early chondrichthyans, *Andreev et al., 2016*). Maxillate placoderms are never recovered as sampled ancestors in the analysis, and the fact that they are of the same age as the osteichthyan *Guiyu* (*Zhu et al., 2009*) precludes this. *Entelognathus* and *Qilinyu* are themselves quite disparate and possess their own specializations, most notably the eyes of *Entelognathus* (*Zhu et al., 2013*; *Zhu et al., 2016*).

The results of our analysis are contingent on a phylogenetic hypothesis, in particular the monophyly of core placoderms, which is only strongly supported under a Bayesian tip-dating approach. The differences between parsimony and Bayesian tip-dated trees are discussed at length in *King et al., 2017*. The hypothesis of placoderm paraphyly (*Brazeau, 2009*; *Davis et al., 2012*; *Zhu et al., 2013*), implies a radically different scenario for character evolution (*Dupret et al., 2014*), in which the maxillate placoderms are not representative of ancestral conditions.

Our study proposes the application of dynamic homology concepts to morphological characters in a Bayesian framework. In this manuscript we have applied the method to placoderm jaw bones, but it could also potentially be used to examine skull roof homologies in the future. It should be noted that the simultaneous analysis of primary and secondary homology has been criticized (*Simmons, 2004*), because adding new morphological characters to a data matrix should be a test of phylogenetic relationships, rather than simply adding further support to a given phylogenetic hypothesis. Thus, it can be argued that multiple conflicting primary homology statements should only be analysed with dynamic homology when they are equally plausible. In such cases, supporting the primary homology statement that best fits a phylogenetic hypothesis is preferable to an arbitrary

choice. There may also exist cases where alternative primary homology statements support different tree topologies, and in this case arbitrary choices of primary homology statements could lead to suboptimal phylogenetic trees.

## Materials and methods

We compiled a morphological data matrix of gnathostome fossils (*Supplementary file 1*). The matrix is based on *King et al., 2017* with a revised taxon and character matrix. The taxon list was updated with the addition of *Gladbachus adentatus*, *Milesacanthus antarctica*, *Nerepisacanthus denisoni*, *Rhinodipterus kimberleyensis*, *Chirodipterus australis*, *Dipterus valenciennesi*, *Tungsenia paradoxa*, *Diplocercides kayseri*, *Qingmenodus yui*, *Raynerius splendens*, *Lehmanosteus hyperboreus*, *Shearsbyaspis oepiki*, and *Qilinyu rostrata*. *Ramirosuarezia boliviana*, *Wuttagoonaspis fletcheri*, *Gavinaspis convergens* and *Osorioichthys marginis* were removed.

Characters concerning the premaxillae, maxillae, dermopalatines and vomers were coded into two alternative *homology alignments*. These characters included presence and absence of these bones, as well as dependent characters. One alignment (homology state 0) was coded according the traditional interpretation of placoderm jaw bones (*Figure 1A*), in which the placoderm supragnathal bones are considered primary homologues of the vomer-dermopalatine series of osteichthyans. A second alignment (homology state 1) was coded according to the alternative interpretation (*Zhu et al., 2016*), in which placoderm supragnathal bones are considered primary homologues of the premaxilla-maxilla series of osteichthyans and maxillate placoderms. In total, the matrix had 489 characters with fixed homology, and 18 with variable homology.

We analysed the matrix in BEAST2.6.2 (*Bouckaert et al., 2019*), using the beagle calculation library (*Ayres et al., 2019*). We used homoplasy-based partitioning (*Rosa et al., 2019*) to account for rate variation among characters. Homoplasy was calculated using an implied weights parsimony analysis in TNT (*Goloboff and Catalano, 2016*), with concavity constant k = 10. Characters with different homoplasy values depending on homology state were assigned the lower value. Characters were partitioned according to the number of states as well as homoplasy. Each partition was assigned a separate mutation rate parameter and was analysed using the Mk substitution model (*Lewis, 2001*). The weighted mean value of the mutation rates was fixed at one, and each individual mutation rate parameter was assigned a normal distribution prior, with mean one and standard deviation 2.

We implemented a sampled ancestor birth-death model (*Gavryushkina et al., 2014*). The birth rate was assigned a lognormal prior with mean (in real space) 0.14 and standard deviation 0.9. Extinction and sampling rates were assigned exponential priors with mean 0.1. Tip dates were assigned to fossil sites with uniform priors on fossil site ages (*King and Rücklin, 2020*). Gnathostomes, gnathostomes+osteostracans and polybranchiaspids were constrained to be monophyletic. The clock model was an uncorrelated lognormal relaxed clock (*Drummond et al., 2006*) with a lognormal prior (mean −5.5, standard deviation 2) on clock rate and an exponential prior (mean 1) on clock standard deviation. We used ancestral sequence logging to reconstruct ancestral states for all characters at the (apomorphy-defined) gnathostome node at every sampled generation of the MCMC. This leads to 1801 'reconstructed ancestors', which comprise a credible set of phenotypes at the gnathostome crown node.

We ran the analysis for 800 million generations, and for four independent runs. The MCMC chain was sampled every 400000 generations, and 10% of the run was discarded as burn-in, resulting in a posterior sample of 1801 trees. Convergence of 4 independent runs was confirmed in Tracer 1.7 (*Rambaut et al., 2018*) and RWTY (*Warren et al., 2017*). Following the recommendations of *O'Reilly and Donoghue, 2018*, we calculated the 50% majority-rule tree in the R package ape (*Paradis and Schliep, 2019*), then time-scaled and annotated this tree using TreeAnnotator 1.10.2 (*Suchard et al., 2018*). The Beast2 xml file is available in the supplementary information (*Supplementary file 2*).

We used distance-based methods to determine the similarity of known fossil taxa to the reconstructed sequences at the gnathostome node. Principal coordinates analysis was performed in the package Claddis (*Lloyd, 2016*) in R 4.0.0 'Arbor Day' (*R Development Core Team, 2018*). We used the Maximum-Observable Rescaled Distance, equivalent to the *Gower, 1971* coefficient for our dataset. First, we performed ordination using the gnathostome fossils in our dataset, and a sample

of the reconstructed ancestors from BEAST2 (*Figure 6A*). This sample consisted of 5% of the posterior sample, from which we excluded those sampled generations where the homology state was 0 (n = 1), for a total of 90 reconstructed ancestors. Homology alignment 1 was used for distance calculations. A second ordination was performed using only placoderms (both core placoderms and maxillate placoderms)(*Figure 6B*). The point cloud of reconstructed ancestors was converted to a density plot using ggplot (*Wickham, 2016*). We also plotted the raw distance measures of each gnathostome taxon to each of the 90 reconstructed ancestors (*Figure 6C*). Finally, we calculated the taxon with the shortest distance to the reconstructed ancestor for the entire posterior distribution (1801 reconstructed ancestors). These calculations used the homology alignment corresponding to the sampled homology state.

## Acknowledgements

We thank John Long and Mike Lee for comments on an earlier version of the manuscript and Min Zhu and Per Ahlberg for reviews. This work is funded by NWO Vidi 864.14.009 to Martin Rücklin.

## Additional information

### Funding

| Funder | Grant reference number | Author |
| --- | --- | --- |
| Nederlandse Organisatie voor Wetenschappelijk Onderzoek | Vidi 864.14.009 | Benedict King Martin Rücklin |

The funders had no role in study design, data collection and interpretation, or the decision to submit the work for publication.

### Author contributions

Benedict King, Conceptualization, Data curation, Software, Formal analysis, Investigation, Methodology, Writing - original draft, Writing - review and editing; Martin Rücklin, Funding acquisition, Writing - review and editing

### Author ORCIDs

Benedict King  https://orcid.org/0000-0002-9489-8274

### Decision letter and Author response

Decision letter https://doi.org/10.7554/eLife.62374.sa1
Author response https://doi.org/10.7554/eLife.62374.sa2

## Additional files

### Supplementary files

• Source data 1. Character state probabilities at the (apomorphy-defined) gnathostome node for all characters.

• Supplementary file 1. Data matrix in nexus format.

• Supplementary file 2. Beast2 xml file.

• Transparent reporting form

### Data availability

The data matrix in nexus format and the BEAST2 xml file are available in the supplementary information. The beast2 source code and R analysis scripts are available at https://github.com/king-ben/homology (copy archived at https://archive.softwareheritage.org/swh:1:rev:6e6dbd77443b0d963640b3cb603c4310b5a4b47e).

The following datasets were generated:

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

## Appendix 1

### Sensitivity analysis

Bayesian tip-dated analysis may be sensitive to incomplete taxon sampling (*O'Reilly and Donoghue, 2020*). The fossil record of early gnathostomes may be biased by the nearshore origination of the major groups (*Sallan et al., 2018*). One example of a possible bias are the antiarchs. The earliest antiarch included in our dataset is the Lochkovian *Yunnanolepis*. However, the antiarch *Shimenolepis* is known from the Silurian (Ludlow) of China, although its fragmentary remains provide few characters for phylogenetic analysis.

To test the effect of Silurian antiarchs on our results we reanalyzed the data with a Ludlow age assigned to *Yunnanolepis*. The major results of the analysis were unchanged, although there was a slight increase in uncertainty. Core placoderm monophyly was supported (pp = 0.98, down from 1.0), with maxillate placoderms as sister group to core placoderms (pp = 0.52, down from 0.70). Homology of arthrodire gnathal plates and the premaxilla/maxilla was supported (pp = 0.98, down from 0.984). Phenetic analysis supported maxillate placoderms as the least diverged known gnathostomes (pp = 0.87, down from 0.95). There was increased support for a member of the core placoderms being the least diverged gnathostome (*Appendix 1—figure 1*), with *Diandongpetalichthys* accounting for most of that probability. Support for key character states at the gnathostome node was slightly reduced (*Appendix 1—table 1*). Overall, this sensitivity shows that our conclusions are robust to at least some issues regarding fossil sampling. However, future studies should aim to further explore the effect of taxon sampling on results.

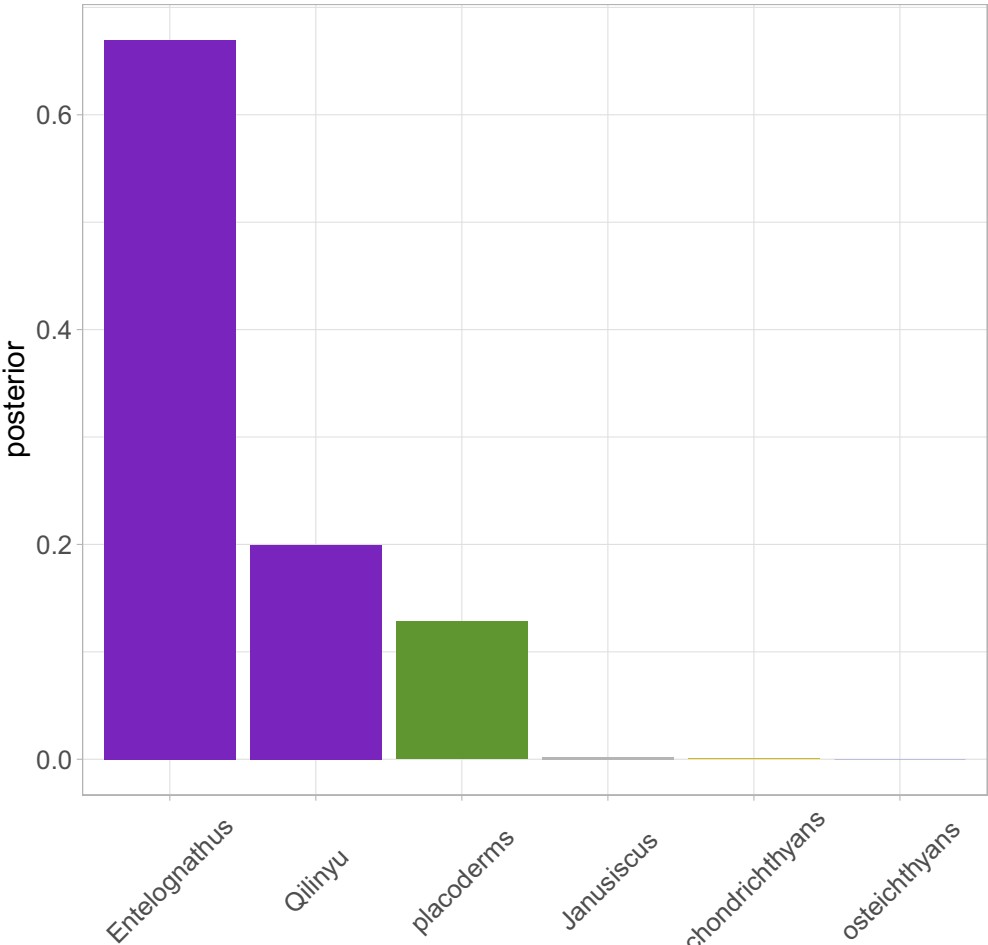

**Appendix 1—figure 1.** Frequency plot of the taxon with the shortest distance to the reconstructed
*Appendix 1—figure 1 continued on next page*

*Appendix 1—figure 1 continued*

ancestor across the whole posterior sample (n = 1801), when data is analysed with a Silurian age for *Yunnanolepis*.

**Appendix 1—table 1.** Probabilities of key character states at the gnathostome node, when data is analysed with a Silurian age for *Yunnanolepis*.

| Character | Reconstructed ancestral state | Posterior probability |
| --- | --- | --- |
| Premaxilla | Present | 0.97 |
| Maxilla | Present | 0.92 |
| Facial laminae | Present | 0.89 |
| Palatal laminae | Present | 0.97 |
| Vomer | Absent | 0.96 |
| Dermopalatine | Absent | 0.88 |
| Nasal capsules | Anterior/ventral | 0.85 |
| Orbit dorsal, surrounded by neurocranium | Absent | 0.96 |
| Claspers | Absent | 0.76 |
| Optic fissure | Absent | 0.69 |

## Appendix 2

### Sources for taxa and age ranges

*Hemicyclaspis murchisoni*

***Stensiö, 1932***
 Shropshire Downtownian. Pridoli, 423–419.2 Ma.

*Cephalaspis lyelli*

***Stensiö, 1932***; ***White, 1958***
 Lower Old Red Sandstone, Glammis. Lochkovian, 419.2–410.8 Ma.

*Zenaspis salweyi*

***Stensiö, 1932***
 Lower Old Red Sandstone. Skirrid Fawr, Senni/St Maughans Formation. Lochkovian, 419.2–410.8 Ma.

*Benneviaspis holtedahli*

***Janvier, 1985a***
 Ben Nevis Formation, Red bBay Group. Late Lochkovian, 413.6–410.8 Ma.

*Boreaspis macrorhynchus*

***Janvier, 1985a***
 Wood Bay Formation. Early Pragian, 410.8–409.7 Ma.

*Norselaspis glacialis*

***Janvier, 1981***
 Wood Bay Formation. Early Pragian, 410.8–409.7 Ma.

*Nectaspis areolate*

***Wängsjö, 1952***, (***Janvier, 1981***)
 Wood Bay Formation. Late Pragian, 408.7–407.6 Ma.

*Procephalaspis oeselensis*

***Robertson, 1939***; ***Denison, 1951***; ***Janvier, 1985b***
 Saaremaa. Ludlow, 427.4–423 Ma.

*Tremataspis mammillata*

***Robertson, 1938a***; ***Robertson, 1938b***; ***Denison, 1947***; ***Denison, 1951***; ***Janvier, 1985b***
 Saaremaa. Ludlow, 427.4–423 Ma.

*Waengsjoeaspis excellens*

***Wängsjö, 1952***; ***Janvier, 1985a***
 Fraenkelryggen Formation. Late Lochkovian, 413.6–410.8 Ma.

*Escuminaspis laticeps*

***Janvier et al., 2004***
 Escuminac Formation. Early Frasnian, 382.7–379.2 Ma.

### Eugaleaspis changi

*Liu, 1965*; *Zhu and Gai, 2007*
Xitun Formation, Liaojaoshan. Late Lochkovian, 413.6–410.8 Ma.

### Hanyangaspis guodingshanensis

*Zhu and Gai, 2007*
Guodingshan Formation. Telychian, 438.5–433.4 Ma.

### Polybranchiaspis liaojiaoshanensis

*Liu, 1965*; *Liu, 1975*
Xishancun and Xitun Formations. Lochkovian, 419.2–410.8 Ma.

### Bannhuanaspis vukhuci

*Janvier et al., 1993*
Bac Bun Formation. Late Lochkovian–early Pragian, 413.6–409.7 Ma.

### Wenshanaspis zhichangensis

*Zhao et al., 2001*
Posongchong Formation, Wenshan. Pragian, 410.8–407.6 Ma.

### Shuyu zhejiangensis

*Gai et al., 2011*
Maoshan Formation. Late Telychian–early Wenlock, 435.1–431.4 Ma.

### Polybranchiaspid sp histological samples

*Wang et al., 2005*
Xishancun and Xitun Formations. Lochkovian, 419.2–410.8 Ma.

### Yunnanolepis sp

*Zhang, 1980*; *Zhu, 1996*
Xishancun and Xitun Formations. Lochkovian, 419.2–410.8 Ma.

### Parayunnanolepis xitunensis

*Zhang et al., 2001*; *Zhu et al., 2012*
Xitun Formation. Late Lochkovian, 413.6–410.8 Ma.

### Microbrachius dicki

*Hemmings, 1978*; *Long et al., 2015*
Eday Flagstone and John O'Groats Sandstone. Lower–middle Givetian, 387.7–384.4 Ma.

### Bothriolepis sp Gogo

*Young, 1984*
Gogo Formation. Early Frasnian, 382.7–379.2 Ma.

### Bothriolepis canadensis

***Downs and Donoghue, 2009***; ***Béchard et al., 2014***
Escuminac Formation. Early Frasnian, 382.7–379.2 Ma.

### Pterichthyodes milleri

***Hemmings, 1978***
Achanarras Horizon. Late Eifelian, 389.6–387.7 Ma.

### Remigolepis walkeri

***Johanson, 1997***
Canowindra. Famennian, 372.2–358.9 Ma.

### Diandongpetalichthys liaojiaoshanensis

***Zhu, 1991***
Xishancun Formation. Lochkovian, 419.2–410.8 Ma.

### Quasipetalichthys haikouensis

***Liu, 1991***
Shixiagou Formation, Ninxia. Givetian, 387.7–382.7 Ma.

### Eurycaraspis incilis

(***Liu, 1991***)
Haikou Formation. Givetian, 387.7–382.7 Ma.

### Lunaspis broili

***Gross, 1961***
Hunsrueck Slate. Late Pragian–early Emsian, 408.7–402.8 Ma.

### Shearsbyaspis oepiki

***Young, 1985***; ***Castiello and Brazeau, 2018***
Taemas-Wee Jasper. Emsian, 407.6–393.3 Ma.

### Macropetalichthys rapheidolabis

***Stensiö, 1925***; ***Stensiö, 1963b***; ***Stensiö, 1969***
Onondaga Limetone. Eifelian, 393.3–387.7 Ma.

### Cowralepis mclachlani

***Ritchie, 2005***; ***Carr et al., 2009***
Merriganowry Shale. Late Givetian–early Frasnian, 384.4–379.2 Ma.

### Sigaspis lepidophora

***Goujet, 1973***
Wood Bay Formation. Early Pragian, 410.8–409.7 Ma.

### Kujdanowiaspis podolica

***Stensiö, 1963a***; ***Dupret, 2010***

Dnister Series, Podolia. Late Lockhovian–lower Pragian, 413.6–409.7 Ma.

### Lehmanosteus hyperboreus

*Goujet, 1984a*

Wood Bay Formation. Early Pragian, 410.8–409.7 Ma.

### Dicksonosteus arcticus

*Goujet, 1975*; *Goujet, 1984b*

Wood Bay Formation. Early Pragian, 410.8–409.7 Ma.

### Groenlandaspis sp Mt Howitt

Specimens listed in *King et al., 2017*

Mt. Howitt. Givetian, 387.7–382.7 Ma.

### Buchanosteus confertituberculatus

*Burrow and Turner, 1998*; *Long et al., 2014*

Buchan. Middle–late Pragian, 409.7–407.6 Ma.

### Parabuchanosteus murrumbidgeensis

*White and Toombs, 1972*; *Young, 1979*; *Burrow and Turner, 1998*

Taemas-Wee Jasper. Emsian, 407.6–393.3 Ma.

### Holonema westolli

*Miles, 1971*

Gogo Formation. Early Frasnian, 382.7–379.2 Ma.

### Coccosteus cuspidatus

*Miles and Westoll, 1968*

Achanarras and Edderton fish bed. Eifelian-Givetian boundary, 394.5–392.1 Ma.

### Incisoscutum ritchiei

*Dennis and Miles, 1981*; *Giles et al., 2013*

Gogo Formation. Early Frasnian, 382.7–379.2 Ma.

### Eastmanosteus calliaspis

*Dennis-Bryan, 1987*

Gogo Formation. Early Frasnian, 382.7–379.2 Ma.

### Compagopiscis croucheri

*Gardiner and Miles, 1994*

Gogo Formation. Early Frasnian, 382.7–379.2 Ma.

### Materpiscis attenboroughi

*Long et al., 2008*; *Trinajstic et al., 2012*

Gogo Formation. Early Frasnian, 382.7–379.2 Ma.

### Austroptyctodus gardineri

*Long, 1997*

Gogo Formation. Early Frasnian, 382.7–379.2 Ma.

### Campbellodus decipiens

*Long, 1997*

Gogo Formation. Early Frasnian, 382.7–379.2 Ma.

### Rhamphodopsis threiplandi

*Miles, 1967*; (*Long, 1997*)

Edderton Fish Beds. Eifelian-Givetian boundary, 394.5–392.1 Ma.

### Brindabellaspis stensioi

*Young, 1980*; *King et al., 2018*

Taemas-Wee Jasper. Emsian, 407.6–393.3 Ma.

### Romundina stellina

*Ørvig, 1975*; *Dupret et al., 2014*; *Dupret et al., 2017*

Prince of Wales Island. Lochkovian, 419.2–410.8 Ma.

### Jagorina pandora

*Stensiö, 1969*; *Young, 1986*

Kellwasserkalk, Bad Wildungen. Late Frasnian, 375.7–372.2 Ma.

### Gemuendina stuertzi

*Gross, 1963*

Hunsrueck Slate. Late Pragian–early Emsian, 408.7–402.8 Ma.

### Entelognathus primordialis

*Zhu et al., 2013*

Kuanti Formation. Ludlow, 427.4–423 Ma.

### Qilinyu rostrata

*Zhu et al., 2016*

Kuanti Formation. Ludlow, 427.4–423 Ma.

### Janusiscus schultzei

*Giles et al., 2015c*

Lower Member, Kureika Formation. Middle Lockhovian, 416.4–413.6 Ma.

### Nerepisacanthus denisoni

*Burrow, 2011*; *Burrow and Rudkin, 2014*

Bertie Formation. Ludlow–Pridoli, 427.4–419.2 Ma.

### Poracanthodes menneri

*Valiukevicius, 1992*

Severnaya Zemlya Formation. Early Lockhovian, 419.2–416.4 Ma.

### Ischnacanthus gracilis
*Watson, 1937*; *Miles, 1973a*; *Burrow et al., 2018*
'Turin Hill'. Lochkovian, 419.2–410.8 Ma.

### Tetanopsyrus lindoei/breviacanthias
*Gagnier et al., 1999*; *Hanke et al., 2001*
MOTH. Lochkovian, 419.2–410.8 Ma.

### Diplacanthus crassisimus
*Watson, 1937*; *Miles, 1973a*; *Burrow et al., 2016*
Moray Firth and Achanarras. Eifelian-Givetian boundary, 394.5–392.1 Ma.

### Milesacanthus antarctica
*Young and Burrow, 2004*
Aztec Siltstone. Givetian, 387.7–382.7 Ma.

### Culmacanthus stewarti
*Long, 1983*
Mt Howitt. Givetian, 387.7–382.7 Ma.

### Euthacanthus macnicoli
*Watson, 1937*; *Miles, 1973a*; *Newman et al., 2014*
'Turin Hill'. Lochkovian, 419.2–410.8 Ma.

### Cassidiceps vermiculatus
*Gagnier and Wilson, 1996*
MOTH. Lockhovian, 419.2–410.8 Ma.

### Promesacanthus eppleri
*Hanke, 2008*
MOTH. Lochkovian, 419.2–410.8 Ma.

### Mesacanthus mitchelli
*Watson, 1937*; *Miles, 1973a*
'Turin Hill' and Farnell. Lochkovian, 419.2–410.8 Ma.

### Cheiracanthus sp
*Watson, 1937*; *Miles, 1973a*
Middle Old Red Sandstone, Moray Firth. Nodular Fish Beds. Eifelian–Givetian, 393.3–382.7 Ma.

### Homalacanthus concinnus
*Gagnier, 1996*
Escuminac Formation. Early Frasnian, 382.7–379.2 Ma.

### Acanthodes bronni

*Gross, 1935*; *Watson, 1937*; *Miles, 1973a*; *Miles, 1973b*; *Coates, 1994*; *Davis et al., 2012*; *Brazeau and de Winter, 2015*
Lebach iIronstone. Asselian, 298.9–293.5 Ma.

### Ptomacanthus anglicus

*Miles, 1973a*; *Brazeau, 2009*; *Brazeau, 2012*; *Dearden et al., 2019*
Wayne Herbert Quarry. Lochkovian, 419.2–410.8 Ma.

### Climatius reticulatus

*Watson, 1937*; *Miles, 1973a*; *Burrow et al., 2015*
'Turin Hill'. Lochkovian, 419.2–410.8 Ma.

### Vernicomacanthus waynensis

*Miles, 1973a*
Wayne Herbert Quarry. Lochkovian, 419.2–410.8 Ma.

### Parexus recurvus

*Watson, 1937*; *Miles, 1973a*; *Burrow et al., 2013*
'Turin Hill'. Lochkovian, 419.2–410.8 Ma.

### Latviacanthus ventspilsensis

*Schultze and Zidek, 1982*
Ventspils. Kemeri stage. Pragian, 410.8–407.6 Ma.

### Brachyacanthus scutiger

*Watson, 1937*
Lower Old Red Sandstone, Farnell. Lochkovian, 419.2–410.8 Ma.

### Brochoadmones milesi

*Hanke and Wilson, 2006*
MOTH. Lochkovian, 419.2–410.8 Ma.

### Gladiobranchus probaton

*Hanke and Davis, 2008*
MOTH. Lochkovian, 419.2–410.8 Ma.

### Kathemacanthus rosulentus

*Gagnier and Wilson, 1996*; *Hanke and Wilson, 2010*
MOTH. Lochkovian, 419.2–410.8 Ma.

### Lupopsyrus pygmaeus

*Hanke and Davis, 2012*
MOTH. Lochkovian, 419.2–410.8 Ma.

### Obtusacanthus corroconis

*Hanke and Wilson, 2004*
MOTH. Lochkovian, 419.2–410.8 Ma.

### Gladbachus adentatus

*Coates et al., 2018*
Lower Plattenkalk. Late Givetian, 382.7–384.4 Ma.

### Cladodoides wildungensis

*Maisey, 2005*
Wildungen Limestone. Late Frasnian, 375.7–372.2 Ma.

### Akmonistion zangerli

*Coates and Sequeira, 1998*; *Coates et al., 1998*; *Coates and Sequeira, 2001*
Manse Burn Formation, Bearsden. Serpukhovian, 330.9–323.2 Ma.

### Cobelodus braincase

*Maisey, 2007*
Fayetteville Formation. Chesterian, 333–318.1 Ma.

### Cladoselache kepleri/fyleri

*Harris, 1938*; *Bendix-Almgreen, 1975*; *Schaeffer, 1981*; *Maisey, 2007*
Cleveland Member of Ohio Shale. Late Famennian, 363.3–358.9 Ma.

### Chondrenchelys problematica

*Moy-Thomas, 1935*; *Finarelli and Coates, 2011*; *Finarelli and Coates, 2014*
Glencartholm Volcanic Beds. Holkerian, 339–337.5 Ma.

### Helodus simplex

*Moy-Thomas, 1936*
Fenton, Staffordshire. Moscovian, 315.2–307 Ma.

### Debeerius ellefseni

*Grogan and Lund, 2000*
Bear Gulch Limestone. Upper Chesterian, 323.1–318.1 Ma.

### Doliodus latispinosus

*Miller et al., 2003*; *Maisey et al., 2009*; *Maisey et al., 2014*; *Maisey et al., 2017*; *Maisey et al., 2018*
'Atholville Beds', Campbellton Formation. Emsian–Eifelian, 407.6–391.4 Ma.

### Hamiltonichthys mapesi

*Maisey, 1989*
Hartford Limetone, Hamilton Quarry. Middle Virgilian, 303.7–298.9 Ma.

### *Onychoselache traquari*

*Dick and Maisey, 1980*; *Coates and Gess, 2007*
Glencartholm Volcanic Beds and Wardie Shales. Holkerian-Asbian, 339–333 Ma.

### *Orthacanthus* sp

*Schaeffer, 1981*
Admiral fFormation. Wolfcampian, 299–280 Ma.

### *Pucapampella rodrigae*

*Maisey, 2001*; *Maisey et al., 2018*
Sica Sica Formation. Eifelian–Givetian, 393.3–382.7 Ma.

### *Tamiobatis vetustus*

*Schaeffer, 1981*; *Williams, 1998*
Cleveland Shale and Salem lLimestone. Famennian, Early Visean, 372.2–358.9 Ma.

### *Tristychius arcuatus*

*Dick, 1978*; *Coates and Gess, 2007*; *Coates and Tietjen, 2018*
Wardie Shales and Manse Burn Formation, Bearsden. Late Visean–lower Serpukhovian, 336.2–328.3 Ma.

### *Dialipina salgueiroensis*

*Schultze, 1968*; *Schultze and Cumbaa, 2001*
Bear Rock Formation. Emsian, 407.6–393.3 Ma.

### *Ligulalepis* braincase

*Basden et al., 2000*; *Basden and Young, 2001*; *Clement et al., 2018*
Taemas-Wee Jasper. Emsian, 407.6–393.3 Ma.

### *Cheirolepis canadensis*

*Pearson and Westoll, 1979*; *Arratia and Cloutier, 1996*
Escuminac Formation. Early Frasnian, 382.7–379.2 Ma.

### *Cheirolepis trailli*

*Pearson and Westoll, 1979*; *Giles et al., 2015a*
Achanarras Limestone, Tynet Burn and Gamrie. Late Eifelian, 389.6–387.7 Ma.

### *Howqualepis rostridens*

*Long, 1988*
Mt. Howitt. Givetian, 387.7–382.7 Ma.

### *Raynerius splendens*

*Giles et al., 2015b*
Upper part of the Grey Member, Ferques Formation. Conodont zone, 373.5–372.5 Ma.

## Mimipiscis toombsi

*Gardiner and Bartram, 1977*; *Gardiner, 1984*; *Giles and Friedman, 2014*
Gogo Formation. Early Frasnian, 382.7–379.2 Ma.

## Moythomasia durgaringa

*Gardiner, 1984*
Gogo Formation. Early Frasnian, 382.7–379.2 Ma.

## Kentuckia deani

*Rayner, 1952*; *Giles and Friedman, 2014*
New Providence Shale Member, Stockdale Formation.
Tournasian-Visean boundary, 347.1–346.3 Ma.

## Meemannia eos

*Zhu et al., 2006*; *Zhu et al., 2010*; *Lu et al., 2016a*
Xitun Formation. Late Lochkovian, 413.6–410.8 Ma.

## Guiyu oneiros

*Zhu et al., 2009*; *Qiao and Zhu, 2010*

## Psarolepis romeri

*Yu, 1998*; *Zhu et al., 1999*; *Zhu and Yu, 2004*; *Zhu and Yu, 2009*
Xishancun Formation. Lochkovian, 419.2–410.8 Ma.

## Achoania jarvikii

*Zhu et al., 2001*; *Zhu and Ahlberg, 2004*; *Zhu and Yu, 2009*
Xitun Formation. Late Lochkovian, 413.6–410.8 Ma.

## Qingmenodus yui

*Lu and Zhu, 2009*; *Lu et al., 2016b*
Posongchong Formation, Wenshan. Pragian, 410.8–407.6 Ma.

## Onychodus jandemarrai

*Andrews et al., 2005*
Gogo Formation, Saddler Formation. Early Frasnian, 382.7–379.2 Ma.

## Miguashaia bureaui

*Cloutier, 1996*; *Forey, 1998*
Escuminac Formation. Early Frasnian, 382.7–379.2 Ma.

## Diplocercides kayseri

*Stensiö, 1922*; *Jarvik, 1980*; *Forey, 1998*
Wildungen Limestone. Late Frasnian, 375.7–372.2 Ma.

## Styloichthys changae

*Zhu and Yu, 2002*; *Friedman, 2007b*

Xitun Formation. Late Lochkovian, 413.6–410.8 Ma.

## Youngolepis praecursor

*Zhang and Yu, 1981*; *Chang, 1982*; *Chang, 1991*; *Chang and Smith, 1992*
Xitun Formation. Late Lochkovian, 413.6–410.8 Ma.

## Powichthys thorsteinssoni

*Jessen, 1975*; *Jessen, 1980*; *Chang and Smith, 1992*
Prince of Wales Island. Late Lockhovian–early Pragian, 413.6–409.7 Ma.

## Diabolepis speratus

*Chang and Yu, 1984*; *Smith and Chang, 1990*; *Chang, 1995*
Xitun Formation. Late Lochkovian, 413.6–410.8 Ma.

## Dipterus valenciennesi

*Parrington, 1950*; *White, 1965*; *Ahlberg and Trewin, 1994*; *Challands, 2015*
Achanarras Limestone. Late Eifelian, 389.6–387.7 Ma.

## Rhinodipterus kimberleyensis

*Clement, 2012*; *Clement and Ahlberg, 2014*
Gogo Formation. Early Frasnian, 382.7–379.2 Ma.

## 'Chirodipterus' australis

*Miles, 1977*; *Henderson and Challands, 2018*
Gogo fFormation. Early Frasnian, 382.7–379.2 Ma.

## Porolepis sp

*Jarvik, 1972*; *Clement, 2004*
Wood Bay Formation. Early Pragian, 410.8–409.7 Ma.

## Glyptolepis groenlandica

*Jarvik, 1972*; *Ahlberg, 1989*
Red Siltstone Member of the Nathorst Fjord group. Late Eifelian–early Givetian, 389.6–386 Ma.

## Tungsenia paradoxa

*Lu et al., 2012*; *Lu et al., 2019*
Posongchong Formation, Wenshan. Pragian, 410.8–407.6 Ma.

## Kenichthys campbelli

*Chang and Zhu, 1993*; *Zhu and Ahlberg, 2004*
Chuangdong Formation. Late Emsian, 398.1–393.3 Ma.

## Osteolepis macrolepidotus

*Westoll, 1936*; *Thomson, 1965*; *Jarvik, 1980*
Tynet Burn. Late Eifelian, 389.6–387.7 Ma.

### Gogonasus andrewsae

*Long, 1985*; *Long et al., 1997*; *Long et al., 2006*; *Holland, 2013*; *Holland, 2014*
Gogo Formation. Early Frasnian, 382.7–379.2 Ma.

### Eusthenopteron foordi

*Jarvik, 1980*; *Porro et al., 2015*
Escuminac Formation. Early Frasnian, 382.7–379.2 Ma.

## Deleted taxa from *King et al., 2017*

### Wuttagoonaspis fletcheri

The description of the braincase of *Wuttagoonaspis* in *Ritchie, 1973* does not match direct observations of specimens (by B. King). Impressions of neurocranial processes can be seen lateral to the sensory grooves in several specimens in the Australia Museum collections. However, since these observations are on unpublished specimens, for reproducibility we elected to remove the taxon.

### Gavinaspis convergens

This taxon is known only from an incomplete skull roof, on which the sutures are unclear. Because of the inability to code almost all characters, this specimen was removed from the matrix. *Lehmanosteus* was added as an alternative example of an early arthrodire.

### Ramirosuarezia boliviana

This is another taxon for which the vast majority of characters cannot be scored. Although the specimen consists of a braincase, almost all neurocranial features are uncertain, even the position of the optic nerve foramen. The inability to score most characters justifies removal of the taxon a priori. In addition, it is also notable that two of the suggested attributions of *Ramirosuarezia*, a decayed rhenanid braincase, or a holocephalan (*Pradel et al., 2009*), receive no support in phylogenetic analyses (e.g. *Coates et al., 2018*). Conversely, an acanthodian identity (deemed 'unlikely' by *Pradel et al., 2009*), receives some support in phylogenetic analysis (*Qiao et al., 2016*).

### Osorioichthys marginis

Based on direct observations of the holotype specimen (by B. King), many of the characters scored from the description were unclear or could not be verified. An important character that influences the position of *Osorioichthys* is the described separation of the posterior nostril and orbit by dermal bone. However, observation of the specimen reveals that this is either an artifact of breakage, or represents the postnasal wall of the neurocranium. *Raynerius* was added as an alternative early acintopterygian with better quality preservation of many features.

## Appendix 3

### Character list
Histology

1. Tessellated calcified cartilage: absent (0); present (1).
*Burrow et al., 2016* c1.

2. Tessellated calcified cartilage: single-layered (0); multi-layered (1).
*Maisey, 2001*, c17.

3. Perichondral bone: present (0); absent (1).
*Donoghue et al., 2000*, c63.

4. Extensive endochondral ossification: absent (0); present (1).
*Zhu et al., 2001* c202; *Brazeau, 2009* c3.

5. Extensive pore canal network: absent (0); present (1).
*Giles et al., 2015a* c8.

6. Three-layered exoskeleton: absent (0); present (1).
*Donoghue et al., 2000*, c71.

7. Cephalic dermoskeleton bone: cellular (0); acellular (1).
*Donoghue et al., 2000* c67; *Sansom, 2009* c73.

8. Perforated horizontal lamina in the sensory canals and vascular system: absent (0); present (1).
*Sansom, 2009* c85.

9. Galeaspidin: absent (0); present (1).
*Sansom, 2009* c87.

10. Dentine: absent (0); present (1).
*Brazeau, 2009* c4.

11. Dentine kind: mesodentine (0); semidentine (1); orthodentine (2).
*Brazeau, 2009* c5.

12. Bone cell lacunae in body scale bases: present (0); absent (1).
*Burrow and Turner, 2010* c61.

13. Main dentinous tissue forming fin spine: osteodentine (0); orthodentine (1).
*Burrow and Turner, 2010* c60.

14. Resorption and redeposition of odontodes: absent or partially developed (0); present (1).
*Zhu et al., 2006* c122.

15. Enamel(oid) present on dermal bones and scales: absent (0); present (1).
*Giles et al., 2015b* c5.

16. Enamel: single-layered (0); multi-layered (1).
*Giles et al., 2015a* c6.

17. Enamel layers: applied directly to one another (ganoine) (0); separated by layers of dentine (1).
*Giles et al., 2015b* c7.

### Braincase proportions

18. Nasal opening(s): dorsal, placed between orbits (0); ventral and anterior to orbits (1).
*Friedman, 2007a* c142.

19. Nasal capsules in anterlolateral corners of orbit: no (0); yes (1).
*King et al., 2017* c96.

20. Elongate preorbital region between orbits and nasal capsules: absent (0); present (1).
*King et al., 2017* c22.

21. Ethmoid region elongate with dorsoventrally deep lateral walls: absent (0); present (1).
*Davis et al., 2012* c73.

22. Tectum orbitale/supraorbital shelf: absent or very narrow (0); present (1).
*Ahlberg and Johanson, 1998b* c83; *King et al., 2017* c102.

23. Supraorbital shelf broad with convex lateral margin: absent (0); present (1).
*Coates and Sequeira, 1998* c17.

24. Orbit dorsal or facing dorsolaterally, surrounded laterally by endocranium: present (0); absent (1).
*Brazeau, 2009* c68.

25. Narrow interorbital septum: absent (0); present (1).
*Brazeau, 2009* c70.

26. Extended prehypophysial portion of sphenoid: absent (0); present (1).
*Brazeau, 2009* c69.

27. Short otico-occipital region of braincase: absent (0); present (1).
*Brazeau, 2009* c74.

28. Parachordal shape: broad, flat (0); keeled with sloping lateral margins (1).
*Brazeau, 2009* c98.

29. Stalk-shaped parachordal/occipital region: absent (0); present (1).
*Giles et al., 2015a* c176.

30. Braincase is series of bilateral ossifications: no (0); yes (1).
*King et al., 2017* c100.

Braincase processes
31. Median rostral dorsal process of braincase: absent (0); present (1).
*King et al., 2017* c101.

32. Rostral processes: absent (0); present (1).
*King et al., 2017* c99.

33. Postnasal wall: absent (0); present (1).
*Clement et al., 2018* c281.

34. Processus supraorbitalis lateralis: absent (0); present (1).
*King et al., 2017* c110.

35. Postorbital process: absent (0); present (1).
*Giles et al., 2015b* c132.

36. Transverse otic process: present (0); absent (1).
*Giles et al., 2015c* c125.

37. Transverse otic process: not extending in front of orbits (0); extending in front of orbits (1).
*Jia et al., 2010* c95.

38. Branchial ridges: present (0); reduced to vagal process (1); absent (articulation made with bare cranial wall) (2).
*Giles et al., 2015c* c166.

39. Vagal process: forked (0); unforked (1).
*Pan et al., 2015* c14; *King et al., 2017* c97.

40. Craniospinal process: absent (0); present (1).
*Giles et al., 2015a* c167.

41. Paravagal fossa: absent (0); present (1).

*Pan et al., 2015* c18.

Braincase fontanelles and fissures

42. Optic fissure: present (0); absent (1).

*Dupret et al., 2014* c255.

43. Ventral cranial fissure: absent (0); present (1).

*Brazeau, 2009* c92.

44. Anterior margin of ventral fissure: straight (0); sinusoidal (1).

*King et al., 2017* c127.

45. Endoskeletal cranial joint: absent (0); present (1).

*Brazeau, 2009* c64.

46. Ventral notch between parachordals: absent (0); present or entirely unfused (1).

*Brazeau, 2009* c97.

47. Parachordal plates: separated from the otic capsule (0); sutured or fused to otic capsule (1).

*Friedman, 2007a* c182.

48. Metotic (otic-occipital) fissure: absent (0); present (1).

*Brazeau, 2009* c93.

49. Occipital arch wedged in between otic capsules: absent (0); present (1).

*Coates and Sequeira, 1998* c9.

50. Precerebral fontanelle: absent (0); present (1).

*Coates and Sequeira, 1998* c21.

51. Anterolateral fenestra in roof of otoccipital: absent (0); present (1).

*King et al., 2017* c111.

52. Hypophysial foramen in braincase: absent (0); present (1).

*King et al., 2017* c114.

53. Posterior dorsal fontanelle: absent (0); present (1).

*Brazeau, 2009* c85.

54. Shape of posterior dorsal fontanelle: approximately as long as broad (0); much longer than wide, slot-shaped (1).

*Coates and Sequeira, 1998* c10.

55. Perilymphatic fenestra: absent (0); present (1).
*Pradel et al., 2011* c16.

56. Vestibular fontanelle: absent (0); present (1).
*Friedman, 2007b* c180.

57. Ventral cranial fissure connects with vestibular fontanelles: no (0); yes (1).
*Coates, 1999* c29; *King et al., 2017* c112.

58. Basal fenestra opening into floor of orbit: absent (0); present (1).
*King et al., 2017* c129.

59. Subpituitary fenestra: absent (0); present (1).
*Goujet and Young, 1995* c12.

60. Basicranial fenestra: absent (0); present (1).
*Ahlberg and Johanson, 1998b* c76.

## Myodomes and articulations
61. Vomeral area with grooves and raised areas: absent (0); present (1).
*Zhu and Schultze, 2001* c142.

62. Ethmoidal articulation of palatoquadrate: absent (0); present (1).
*King et al., 2017* c122.

63. Ethmoid articulation for palatoquadrate: extends posteriorly to the level of N.II (0); placed on postnasal wall (1); majority of facet anterior to postnasal wall (2).
*Friedman, 2007a* c172.

64. Eye stalk or unfinished area on neurocranial wall for eye stalk: absent (0); present (1).
*Zhu and Schultze, 2001* c36.

65. Position of basal/basipterygoid articulation: same anteroposterior level as hypophysial opening (0); anterior to hypophysial opening (1).
*Brazeau, 2009* c81.

66. Basipterygoid process (basal articulation) with vertically oriented component: absent (0); present (1).
*Brazeau, 2009* c83.

67. Expanded articular area anterior to basipterygoid process: absent (0); present (1).
*King et al., 2017* c103.

68. Orbital/palatobasal articulation: posterior to optic foramen (0); anterior to optic foramen (1).

*King et al., 2017* c123.

69. Descending process of sphenoid (with its posterior extremity lacking periostegeal lining): absent (0); present (1).

*Ahlberg, 1991* c53.

70. Processus connectens: short (0); elongate (1).

*Lu et al., 2016a* c66.

71. Articulation between neurocanium and palatoquadrate posterodorsal to orbit (suprapterygoid articulation): absent (0); present (1).

*Giles et al., 2015a* c147.

72. Division of postorbital palatoquadrate articulation into dorsal and ventral components: absent (0); present (1).

New, adapted from *King et al., 2017* c128. Two condyles in the dorsal otic region are found in *Acanthodes* and *Homalacanthus*.

73. Periotic process: absent (0); present (1).

*Pradel et al., 2011* c11.

74. Hyomandibula articulating with braincase: yes (0); no (1).

*King et al., 2017* c121.

75. Position of hyomandibular articulation on neurocranium: Anterior position, suborbital (0); posterior position, behind orbit (1).

*Brazeau, 2009* c89; *King et al., 2017* c369.

76. Relative position of jugular groove and hyomandibular articulation: hyomandibula dorsal or same level (0); jugular vein passing dorsal or lateral to hyomandibula (1).

*Brazeau and de Winter, 2015* c237.

77. Articulation facet with hyomandibular: single-headed (0); double-headed (1).

*Zhu and Schultze, 2001* c128.

78. Hyomandibular facets where they straddle the jugular canal: narrowly separated (0); broadly separated (1).

*Friedman, 2007b* c176.

79. Articulation surface on ventrolateral surface of otic capsule: absent (0); present (1).

New. There is some uncertainty what articulation surfaces on the otic capsule wall are for (see e.g. *Chang, 1982*), but they are most commonly assumed to be for the articulation of an infrapharyngo-branchial. This character simply codes the presence or absence of an articulation.

80. Paired occipital facets: absent (0); present (1).

*Giles et al., 2015b* c177.

81. Position of myodome for superior oblique eye muscles: posterior and dorsal to foramen for nerve II (0); anterior and dorsal to foramen for nerve II (1).

*Brazeau, 2009* c63.

82. Medial recess of the posteroventral myodome: absent (0); present (1).

*Sansom, 2009* c103.

## Braincase ridges

83. Pronounced sub-ethmoidal keel: absent (0); present (1).

*Coates and Sequeira, 1998* c22; *Brazeau, 2009* c62.

84. Subcranial ridges: absent (0); present (1).

*Giles et al., 2015a* c141.

85. Dorsal ridge: absent (0); present (1).

*Coates and Sequeira, 1998* c11; *Davis et al., 2012* c91. Includes the unpaired median crista of lungfishes.

86. Shape of median dorsal ridge anterior to endolymphatic fossa: developed as a squared-off ridge or otherwise ungrooved (0); bears a midline groove (1).

*Giles et al., 2015a* c161.

87. Dorsal otic ridge forming a horizontal crest: absent (0); present (1).

*Coates and Sequeira, 1998* c11; *Pradel et al., 2011* c12.

88. Hypotic lamina (and dorsally directed glossopharyngeal canal): absent (0); present (1).

*Davis et al., 2012* c103.

89. Dorsolateral cristae: absent (0); present (1).

New.

90. Dorsolateral cristae fenestrated: no (0); yes (1).

*Friedman, 2007a* c16.

91. Lateral (parotic) cristae: absent (0); present (1).

New. The parotic crista of sarcopterygians and the lateral cristae of lungfishes are here considered potential homologues following *Miles, 1977*. *Friedman, 2007b* compares the dorsolateral cristae as homologous to the parotic cristae, but cites Miles, so this is assumed to be erroneous.

## Notochord

92. Size of aperture to notochordal canal: much smaller than foramen magnum (0); as large, or larger, than foramen magnum (1).

*Giles et al., 2015a* c178.

93. Notochord short and stopped by occipital cotylus: absent (0); present (1).

*Pradel et al., 2011* c21.

94. Unconstricted cranial notochord: absent (0); present (1).

*Ahlberg and Johanson, 1998b* c153.

## Spiracle

95. Spiracular groove on basicranial surface: absent (0); present (1).

*Brazeau, 2009* c65.

96. Spiracular groove on lateral commissure: absent (0); present (1).

*Davis et al., 2012* c63.

97. Endoskeletal spiracular enclosure: absent (0); present (1); spiracular canal (2).

*Clement et al., 2018* c267.

## Endocast

98. Relationship of cranial endocavity to basisphenoid: endocavity occupies full depth of sphenoid (0); endocavity dorsally restricted (1).

*Giles et al., 2015b* c140.

99. Nasal sacs: unpaired (0); paired (1).

*King et al., 2017* c130.

100. Olfactory tracts: separate tracts (0); single anterior division of the cranial cavity (1).

New. In coelacanths there are no separate canals for the olfactory nerves but rather a large anterior division of the endocavity through which the olfactory tracts pass.

101. Rostral organ: absent (0); present (1).

*Friedman, 2007b* c145.

102. Hypophyseal chamber: projects posteroventrally (0); projects ventrally or anteroventrally (1).

*Clement et al., 2018* c270.

103. Paired recesses anterior of hypophysial fossa: absent (0); present, blind (1); present, connect via canals to cranial cavity (2).

New. *Tungsenia* has paired recesses at the anterior of the hypophysial recess (*Lu et al., 2012* fig. 4b ?PT), interpreted as similar to the pars tuberalis of living urodeles. In *Glyptolepis* there are also extensions of the hypophysial recess that connect anteriorly to the cranial cavity (*Jarvik, 1972* fig. 17A c.p.tub).

104. Optic lobes: narrower than cerebellum (0); same width or wider than cerebellum (1).

*Lu et al., 2017* c271.

105. Lateral cranial canal: absent (0); present (1).

*Coates, 1999* c32.

106. Supraotic cavity: absent (0); present (1).

*Lu et al., 2017* c275.

107. Endolymphatic ducts: posteriodorsally angled tubes (0); tubes oriented vertically through median endolymphatic fossa (1).

*Coates and Sequeira, 2001* c73; *Brazeau, 2009* c87.

### Inner ear

108. Labyrinth cavity: separated from the main neurocranial cavity by a cartilaginous or ossified capsular wall (0); skeletal capsular wall absent (1).

*Davis et al., 2012* c82.

109. External (horizontal) semicircular canal: absent (0); present (1).

*Brazeau, 2009* c83.

110. External (horizontal) semicircular canal: joins the vestibular region dorsal to posterior ampulla (0); joins level with posterior ampulla (1).

*Davis et al., 2012* c87.

111. Horizontal semicircular canal in dorsal view: medial to path of jugular vein (0); dorsal to jugular vein (1).

*Giles et al., 2015b* c154.

112. Horizontal semicircular canal: horizontally orientated (0); obliquely orientated (1).

*Lu et al., 2017* c274.

113. Crus commune: dorsal to endocranial roof (0); ventral to endocranial roof (1).

*Lu et al., 2017* c272.

114. Sinus superior: absent or indistinguishable from union of anterior and posterior canals with saccular chamber (0); present (1).

*Davis et al., 2012* c86.

115. Utricular recess: absent (0); present small (1); present large (2).

New. A diverticulum of the labyrinth cavity at the junction of the external semicircular canal and the sacculus, interpreted as housing the utriculus. See *Brazeau and Friedman, 2014* p18 for discussion.

116. Lagenar recess: absent (0); present (1).

New. A large recess at the posterior end of the saccular chamber for the lagena is well developed in *Diplocercides* (*Jarvik, 1980*, fig.217 space for lagena). Recently, a similar recess was described for the chondrichthyan *Tristychius* (*Coates et al., 2018*, fig.11).

117. Number of SEL canals: five (0); less than 5 (1).

*Sansom, 2009* c91.

118. SEL one canal bifurcation: between orbit and lateral field (0); close to field (1); close to orbit (2).

*Sansom, 2009* c110; *King et al., 2017* c92.

### Blood vessels

119. Canal for efferent pseudobranchial artery within basicranial cartilage: absent (0); present (1).

*Brazeau, 2009* c80.

120. Entrance of internal carotids in 'tropibasic' braincase: through basisphenoid pillar (0); through orbits (1).

New. Revised from *King et al., 2017* c38 and c125 to remove redundancy. This character is only applicable to taxa with a 'tropibasic' braincase: i.e. it is dependent on character 98.

121. Entrance of internal carotids: through separate openings flanking the hypophyseal opening or recess (0); through a common opening at the central midline of the basicranium (1).

*Brazeau, 2009* c79.

122. Canal for lateral dorsal aorta within basicranial cartilage: absent (0); present (1).

*Coates and Sequeira, 1998* c4; *Brazeau, 2009* c78.

123. Midline canal in basicranium for dorsal aorta: absent (0); present (1).

*Zhu et al., 2013* c234.

124. Jugular canal: long (invested in otic region along length of skeletal labyrinth) (0); short (restricted to region anterior of skeletal labyrinth) (1); absent (jugular vein uninvested in otic region) (2).

*Giles et al., 2015b* c126.

125. Canal for jugular in postorbital process: absent (0); present (1).

*Giles et al., 2015c* c133.

126. Pituitary vein canal: dorsal to level of basipterygoid process (0); flanked posteriorly by basipterygoid process (1).

*Davis et al., 2012* c84.

127. Pituitary vein canal: Discontinuous, enters cranial cavity (0); Discontinuous, enters hypophysial recess (1); Continuous transverse canal (2).

*Clement et al., 2018* c282.

128. Marginal vein: absent (0); present (1).

*Sansom, 2009* c93.

## Cranial nerves

129. Olfactory tracts: short, with olfactory capsules situated close to telencephalon cavity (0); elongate and tubular (much longer than wide) (1).

*Brazeau, 2009* c60.

130. Rostral tubuli: absent (0); present (1).

*Cloutier and Ahlberg, 1996* c77.

131. Profundus and trigeminal nerves: emerge from cranial cavity separately (0); emerge from cranial cavity together (1).

*King et al., 2017* c94.

132. Size of anterior profundus canal: small (0); large (1).

*Zhu and Schultze, 2001* c144

Definition revised to remove the postnasal wall, as an anterior profundus foramen can sill be present when the posnasal wall is poorly developed, in particular in '*Ligulalepis*'.

133. Series of perforations for innervation of supraorbital sensory canal in supraorbital shelf: absent (0); present (1).

*Giles et al., 2015c* c134.

134. Profundus nerve enters orbit with jugular vein: no (0); yes (1).

New. In general, the profundus nerve enters the orbit through a foramen on the posteriodorsal wall. However, in '*Chirodipterus*' the profundus nerve first joins the jugular vein canal (*Henderson and Challands, 2018* p16).

135. Relative position of trigeminal nerve: behind endoskeletal cranial joint (0); through endoskeletal cranial joint (1); anterior to endoskeletal joint (2).

*Zhu and Schultze, 2001* c134. This character is here adapted to include an extra state in when the intracranial joint is behind the trigeminal nerve. It is dependent on the presence of an intracranial joint.

136. Palatine nerve canal: absent (0); present (1).

New. A palatine nerve canal is present in gnathostomes, but is apparently unknown in osteostracans or galeaspids (see discussion in *King et al., 2017*)

137. Hyoid ramus of facial nerve (N. VII) exits through posterior jugular opening: absent (0); present (1).

*Friedman, 2007b* c179.

138. Glossopharyngeal nerve (N. IX) exit: foramen situated posteroventral to otic capsule and anterior to metotic fissure (0); through metotic fissure (1).

*Coates and Sequeira, 1998* c2; *Brazeau, 2009* c73.

139. Spino-occipital nerve foramina: two or more, aligned horizontally (0); one or two, dorsoventrally offset (1).

*Coates and Sequeira, 1998* c8; *Brazeau, 2009* c95.

### Dermal palate bones

140. Median dermal bone of palate (parasphenoid): absent (0); present (1).
*Brazeau, 2009* c57.

141. Ascending process of parasphenoid: absent (0); present (1).
*Patterson, 1982a* c9.

142. Shape of parasphenoid denticulated field: broad rhomboid or lozenge-shaped (0); broad, splint-shaped (1); slender, splint-shaped (2).
*Friedman, 2007b* c168.

143. Parasphenoid denticulated field with multifid anterior margin: absent (0); present (1).
*Friedman, 2007b* c167.

144. Parasphenoid: protruding forward into ethmoid region of endocranium (0); behind ethmoid region (1).
*Zhu and Schultze, 2001* c124.

145. Denticulated field of parasphenoid: without spiracular groove (0); with spiracular groove (1).
*Friedman, 2007b* c82.

146. Parasphenoid denticle field: terminates at or anterior to level of foramina for internal carotid arteries (0); extends posterior to foramina for internal carotid arteries (1).
*Friedman, 2007b* c170.

147. Four carotid foramina in parasphenoid: absent (0); present (1).
*Lu et al., 2012* c98; *King et al., 2017* c138.

148. Buccohypophysial canal in parasphenoid: single (0); paired (1).
*Giles et al., 2015a* c114.

149. Posterior stalk of parasphenoid covering otic region: absent (0); present (1).
*Friedman, 2007a* c63.

150. Parasphenoid posterior stalk furrow: absent (0); present (1).
*Schultze, 2001* c51.

151. Prespiracular dental plate: absent (0); present (1).
*King et al., 2017* c108.

152. Parotic dental plate: absent (0); present (1).
*King et al., 2017* c139.

## Skull roof, overall features

153. Dermal skull roof: includes large dermal plates (0); consists of undifferentiated plates or tesserae (1).

*Brazeau, 2009* c19.

154. Tesserae morphology: large interlocking polygonal plates (0); microsquamose, not larger than body tesserae (1).

*Brazeau, 2009* c20.

155. Extent of dermatocranial cover: complete (0); incomplete (scale-free cheek and elsewhere) (1).

*Brazeau, 2009* c21.

156. Series of paired median skull roofing bones that meet at the dorsal midline of the skull (rectilinear skull roof pattern): absent (0); present (1).

*Brazeau, 2009* c24.

157. Dermal intracranial joint: absent (0); present (1).

*Cloutier and Ahlberg, 1996* c81.

158. Cranial spines: absent (0); present, multicuspid (1); present, monocuspid (2).

*Giles et al., 2015c* c36.

## Skull roof, foramina

159. Pineal opening perforation in dermal skull roof: absent (0); present (1).

*Brazeau, 2009* c26.

160. Location of pineal foramen/eminence: level with posterior margin of orbits (0); well posterior of orbits (1).

*Ahlberg and Johanson, 1998b* c37.

161. Endolymphatic ducts open in dermal skull roof: present (0); absent (1).

*Brazeau, 2009* c22.

162. Endolymphatic ducts with oblique course through dermal skull bones: absent (0); present (1).

*Goujet and Young, 1995* c8.

163. Endolymphatic duct relationship to median skull roof bone (i.e. nuchal plate): within median bone (0); on bones flanking the median bone (e.g. paranuchals) (1).

*Giles et al., 2015c* c40.

164. Dermal plate associated with pineal eminence or foramen: contributes to orbital margin (0); plate bordered laterally by skull roofing bones (1).

*Giles et al., 2015c* c42.

## Skull roof, snout

165. Median rostral extension of head shield: absent (0); present (1).

*Sansom, 2009* c1.

166. Tooth-bearing median rostral: absent (0); present (1).

*Cloutier and Ahlberg, 1996* c22.

167. T-shaped rostral: absent (0); present (1).

*Carr and Hlavin, 2010* c5; *King et al., 2017* c237.

168. Multiple postrostral bones: no (0); yes (1).

New. Homology of snout bones (i.e. the bones anterior to the parietals) across gnathostomes are difficult to assess. This character simply makes the distinction between the mosaic of small irregular bones (postrostrals, nasals, tectals) found in sarcopterygians with the relatively small number of larger plates in actinopterygians and placoderms.

169. Number of median bones anterior to parietals: none (0); one (1); two (2).

New. This character reformulates a number of previous characters regarding the presence of rostral and premedian plates. In placoderms the first median bone anterior to the patietals (preorbitals) is generally termed the rostral, while the second is called the premedian or internasal. In osteichthyans they are termed as the postrostral and rostral. Here we remove the position of the nasal capsules from the definition of a premedian plate (e.g. Zhu et al. c148) as the position of the nasal capsules is dealt with in other characters. In taxa with a rostral mosaic of bones (character 168), this character is considered inapplicable.

170. Premedian plate: large plate (0); reduced to internasal plate (1).

*Zhu et al., 2016* c157, revised. This character is contingent on the presence of a premedian plate. This is covered in character 169 state 2.

171. Paired prenostril trenches on premedian plate: absent (0); present (1).

New. Paired prenostril trenches are present on the premedian/internasal plate of Qilinyu. This character is contingent on the presence of a premedian plate (Character 169 state 2).

172. Unornamented shelf and rostrocaudal groove on premedian: absent (0); present (1).

*Jia et al., 2010* c3.

173. Preorbital depression: absent (0); present (1).

*Jia et al., 2010* c6.

174. Supraorbital: absent (0); present (1).

*Cloutier and Ahlberg, 1996* c28.

175. Lateral plate: absent (0); present (1).

*Zhu et al., 2013* c157.

176. Prelateral plate: absent (0); present (1).

*King et al., 2017* c251.

177. Submarginal articulation: absent (0); present (1).

*Jia et al., 2010* c16.

178. Parietals (preorbitals of placoderms) surround pineal foramentotoeminence: yes (0); no (1).

*Ahlberg and Johanson, 1998b* c38.

179. Median bone separating parietals: absent (0); present (1); present and separates postparietals as well (2).

*King et al., 2017* c271, revised.

180. Paraorbital plate separating suborbital from orbit: absent (0); present (1).

*King et al., 2017* c253.

## Skull roof, sclerotic ring

181. Sclerotic ring: absent (0); present (1).

*Giles et al., 2015a* c52.

182. Number of sclerotic plates: four or less (0); more than four (1).

*Cloutier and Ahlberg, 1996* c49.

183. Sclerotic ring incorporated into skull: no (0); yes (1).

*King et al., 2017* c244.

## Skull roof, back half

184. Dermal bone (sarcopt postorbital) between jugal and intertemporal: absent (0); present (1).

*King et al., 2017* c279.

185. Complete enclosure of spiracle by skull roof bones: absent (0); present (1).

*Friedman, 2007b* c148.

186. Suture between paired skull roofing bones (centrals of placoderms postparietals of osteichthyans): straight (0); sinusoidal (1).

*Giles et al., 2015a* c49.

187. Number of bones bearing otic canal between dermosphenotic and lateral extrascapular: one (0); two (1); more than two (2).

New. These bones are termed the marginal and anterior paranuchal in placoderms and the supratemporal and tabular in osteichthyans.

188. Supratemporal contact with postparietal: present (0); absent due to anterior displacement (1); absent due to lateral displacement (2).

*Swartz, 2009* c15; *King et al., 2017* c273.

189. Supratemporal contact with nasal: (0); present (1).

*Gardiner and Schaeffer, 1989* c26.

190. Contact of tabular or anterior paranuchal with postparietal or central: less than half length of postparietal (0); extends most of the length of postparietal (1).

New.

191. Series of bones lateral to supratemporal series: absent (0); single bone (1); two bones (2).

*King et al., 2017* c263.

192. Number of extrascapulars: uneven (0); paired (1).

*Cloutier and Ahlberg, 1996* c40.

193. Number of paired extrascapulars: one pair (0); two pairs (1).

*Gardiner and Schaeffer, 1989* cA8.

194. Medial processes of paranuchal wrapping posterolateral corners of nuchal plate: absent (0); present (1); paranuchals precluded from nuchal by centrals (2).

*Giles et al., 2015a* c50. The fourth state of the Giles et al character is not included here (and the taxa are considered inapplicable) as the presence of a median posterior skull roof bone is dealt with by character 192.

195. Nuchal plate: without orbital facets (0); with orbital facets (1).

*Jia et al., 2010* c14.

196. Centronuchal plate: absent (0); present (1).

*Dupret et al., 2009* c17.

197. Contact of nuchal or centronuchal plate with paired preorbital plates: absent (0); present (1).

*Zhu et al., 2013* c164.

198. Postnuchal plate: absent (0); present (1).

*Dupret et al., 2009* c45; *King et al., 2017* c239.

199. Presupracleithrum: absent (0); present (1).

*Patterson, 1982b* c13.

200. Fused scale rows on head shield: absent (0); present (1).

*Sansom, 2009* c43.

201. Dorsal spinal process on head shield: absent (0); present (1).

*Sansom, 2009* c44.

202. Cornual extensions: absent (0); present (1).
*Sansom, 2009* c36; *Zhu and Gai, 2007* c14.

203. Spines on cornual extension: absent (0); present (1).
*Zhu and Gai, 2007* c18.

204. Most posterior bones flanking postparietals: level with posterior margin of postparietals (0); extend posterior to posterior margin of potparietals (1).
*Lu et al., 2016b* c238.

## Skull roof, joint
205. Type of dermal neck-joint: overlap (0); ginglymoid (1).
*Zhu et al., 2013* c169; *Giles et al., 2015a* c60.

206. Type of ginglymoid neck-joint: conventional (0); reverse (1).
*King et al., 2017* c174.

207. Dermal neck-joint between paired main-lateral-line-bearing bones of skull and shoulder girdle: absent (0); present (1).
*Zhu et al., 2013* c168.

## Skull roof, visceral
208. Broad supraorbital vaults: absent (0); present (1).
*Giles et al., 2015a* c44.

209. Paired pits on ventral surface of nuchal plate: absent (0); present (1).
*Giles et al., 2015b* c51.

210. Preorbital recess: absent (0); present (1).
*Jia et al., 2010* c8; *King et al., 2017* c247.

211. Preorbital recess: restricted to premedian plate (0); extends to lateral plates (1).
*Jia et al., 2010* c8; *King et al., 2017* c248.

212. Posterior descending lamina of skull roof: absent (0); present (1).
*Pan et al., 2015* c6.

213. Mesial lamina of marginal plate: absent (0); present (1).
*King et al., 2017* c254.

## Skull roof, fields
214. Lateral fields: absent (0); present (1).
*Sansom, 2009* c4.

215. Division of lateral fields: absent (0); divided once (1); divided twice (2).
*Sansom, 2009* c5-6; *King et al., 2017* c220.

216. Lateral fields extend posterior to pectoral sinus: absent (0); present (1).
*Sansom, 2009* c10.

217. Lateral field extends to cornua: no (0); yes (1).
*Sansom, 2009* c11.

218. Median field: absent (0); present (1).
*Sansom, 2009* c13.

219. Median field separation from pineal plate or foramen: absent (0); present (1).
*Sansom, 2009* c15.

220. External endolymphatic duct opens within median field: internal (0); external (1).
*Sansom, 2009* c17.

221. Median dorsal opening: absent (0); present (1).
*Donoghue et al., 2000* c14; *Zhu and Gai, 2007* c1.

222. Shape of median dorsal opening: transverse slit-like (0); oval-like (1); slender longitudinal oval (2).
*Zhu and Gai, 2007* c6.

Nostrils

223. External nasal opening: single median (0); paired (1).
*Donoghue et al., 2000* c14; *Sansom, 2009* c25.

224. Nostrils enclosed in dermal skull roof: yes (0); no (1).
*King et al., 2017* c256.

225. Nasohypophysial opening shape: unconstricted (0); constricted (1); split (2).
*Sansom, 2009* c228; *King et al., 2017* c228.

226. Posterior nostril: associated with orbit (0); not associated with orbit (1).
*Cloutier and Ahlberg, 1996* c46.

227. Dermintermedial process: absent (0); present (1).
*Zhu and Schultze, 2001* c37.

228. Position of posterior nostril: external, far from jaw margin (0); external, close to jaw margin (1); palatal (2).
*Zhu and Schultze, 2001* c39.

229. Lacrimal posteriorly enclosing posterior nostril: absent (0); present (1).
*Zhu and Schultze, 2001* c28.

230. Premaxilla contributes to posterior nostril: absent (0); present (1).
*Friedman and Blom, 2006* c7.

## Operculogular

231. Opercular cover of branchial chamber: complete or partial (0); separate gill covers and gill slits (1).
*Davis et al., 2012* c32.

232. Branchiostegals: absent (0); present (1).
*Brazeau, 2009* c31.

233. Branchiostegal plate series along ventral margin of lower jaw: absent (0); present (1).
*Brazeau, 2009* c32.

234. Branchiostegal ossifications: plate-like (0); narrow and ribbon-like (1).
*Brazeau, 2009* c33; *Davis et al., 2012* c29.

235. Branchiostegal ossifications: ornamented (0); unornamented (1).
*Brazeau, 2009* c34.

236. Imbricated branchiostegal ossifications: absent (0); present (1).
*Brazeau, 2009* c35.

237. Shape of opercular (submarginal) ossification: broad plate that tapers towards its proximal end (0); narrow, rod-shaped (1).
*Brazeau, 2009* c37.

238. Size of lateral gular plates: extending most of length of the lower jaw (0); restricted to the anterior third of the jaw (no longer than the width of three or four branchiostegals (1)).
*Brazeau, 2009* c39.

239. Median gular: present (0); absent (1).
*Cloutier and Ahlberg, 1996* c66.

240. Number of branchiostegal rays per side: 10 or more (0); 2–7 (1); one (2).
*Cloutier and Ahlberg, 1996* c63.

241. Accessory operculum: absent (0); present (1).
*Dietze, 2000* c56.

242. Oralobranchial covering: tesserae (0); plates (1).
*Sansom, 2009* c60; *King et al., 2017* c232.

243. Headshield enclosed posteriorly behind oralobranchial chamber: no (0); yes (1).
*King et al., 2017* c235.

## Cheek

244. Cheek plate: undivided (0); divided (i.e. squamosal and preopercular) (1).

*Giles et al., 2015a* c54.

245. Subsquamosals in taxa with divided cheek: absent (0); present (1).

*Zhu and Schultze, 2001* c64; *Giles et al., 2015b* c54.

246. Preopercular shape: rhombic (0); bar-shaped (1).

*Zhu and Schultze, 2001* c71.

247. Consolidated cheek plates: absent (0); present (1).

*Brazeau, 2009* c25.

248. Enlarged postorbital tessera separate from orbital series: absent (0); present (1).

*Brazeau, 2009* c30.

249. Most posterior major bone of cheek bearing preopercular canal (preopercular) extending forward, close to orbit: absent (0); present (1).

*Zhu and Schultze, 2001* c58; *Zhu et al., 2009* c59.

250. Contact between most posterior major bone of cheek bearing preopercular canal and maxilla: present (0); absent (1).

*Zhu and Schultze, 2001* c66.

251. Bone bearing both quadratojugal pit-line and preopercular canal: absent (0); present (1).

*Friedman, 2007b* c42.

252. Notch in anterior margin of jugal: absent (0); present (1).

*Cloutier and Arratia, 2004* c81.

253. Quadratojugal: present (0); absent (1).

*Zhu and Schultze, 2001* c57; *Dietze, 2000* c31.

254. Lacrimal: absent (0); present (1).

*King et al., 2017* c257.

## Premaxilla/maxilla

255. Premaxilla: extends under orbit (0); restricted anterior to orbit (1).

*Friedman, 2007b* c150.

256. Premaxillae with inturned symphysial processes: absent (0); present (1).

*Friedman, 2007b* c149.

257. Premaxilla forming part of orbit: absent (0); present (1).

*Cloutier and Arratia, 2004* c18.

258. Posterior expansion of maxilla (maxilla cleaver-shaped): present (0); absent (1).

*Lund et al., 1995* c52; *Zhu and Schultze, 2001* c54.

259. Contribution by maxilla to posterior margin of cheek: present (0); absent (1).

*Friedman, 2007b* c151.

260. Ventral margin of maxilla: straight (0); curved (1).

*Dietze, 2000* c26.

261. Orbital process of maxilla: absent (0); present (1).

*King et al., 2017* c282.

## Dermal ornament and pores

262. Dermal ornamentation: smooth (0); ridges (1); tuberculate (2).

*Giles et al., 2015b* c29; *King et al., 2017* c205.

263. Size of cosmine pores: small (0); large (1).

*Zhu et al., 2001* c149.

264. Pore clusters: absent (0); present (1).

*Zhu and Schultze, 2001* c207.

265. Westoll-lines: absent (0); present (1).

*Zhu and Schultze, 2001* c207.

266. Transverse external groove behind pineal opening: absent (0); present (1).

*King et al., 2017* c255.

267. Sensory foramina on skull roof, behind orbits: absent (0); present (1).

*King et al., 2017* c241.

268. Cutaneous pits on cheek bones: absent (0); present (1).

New. Here we combine previous characters dealing with sensory pits on the cheeks of osteichthyans (*Ahlberg and Johanson, 1998b* 63) and placoderms (*King et al., 2017* 240, 241). There is no a priori reason to reject homology of these pits.

## Teeth

269. Oral dermal tubercles borne on jaw cartilages: absent (0); present (1).

*Brazeau, 2009* c41.

270. Oral tubercles in patterned rows (teeth): absent (0); present (1).

*Brazeau, 2009* c42.

271. Basal resorption of teeth: absent (0); present (1).

New. This character can be scored based on the presence of replacement pits.

272. Oral dermal tubercles fused to jaw cartilages: absent (0); present (1).

New. *Pucapampella* has the unusual condition in which statodont teeth are fused directly to the jaw cartilage, which extended into the core of the teeth (*Maisey et al., 2018* p99).

273. Tooth whorls: absent (0); present (1).

*Brazeau, 2009* c43.

274. Tooth whorls extent: at symphysis (0); along entire jaw (1).

*Giles et al., 2015b* c83.

275. Distribution of marginal tooth whorls: upper and lower jaws (0); lower jaws only (1); upper jaws only (2).

*Giles et al., 2015b* c84.

276. Bases of marginal tooth whorls: single, continuous plate (0); some or all whorls consist of separate tooth units (1).

*Brazeau, 2009* c44; *Davis et al., 2012* c41.

277. Toothplates: absent (0); present (1).

*Coates et al., 2018* c85.

278. Extramandibular dentition: absent (0); present (1).

*Lu et al., 2012* c392.

279. Number of tooth rows on outer dental arcade: one (0); two (1).

*Friedman, 2007a* c157; *King et al., 2017* c380.

280. Teeth of dentary: reaching anterior end of dentary (0); not reaching anterior end (1).

*Ahlberg and Johanson, 1998b* c11.

281. Fangs of coronoids (sensu stricto): absent (0); present (1).

*Ahlberg et al., 2000* c15.

282. Number of fang pairs on posterior coronoid: one (0); two (1); none (2).

*Ahlberg and Johanson, 1998b* c13.

283. Marginal denticle band on coronoids: broad band, at least posteriorly (0); narrow band with 2–4 denticle rows (1).

*Ahlberg and Johanson, 1998b* c9.

284. Teeth radial rows on prearticular: absent (0); present (1).

*Zhu and Schultze, 2001* c95.

285. Core of oral dermal tubercles: open pulp cavity (0); vascular network (osteodentine) (1).

New. We do not differentiate between the absence of a pulp cavity and a secondarily infilled pulp cavity.

286. Enamel(oid) on teeth: absent (0); present (1).

*Giles et al., 2015a* c79.

287. Plicidentine: absent (0); present (1).

*Cloutier and Ahlberg, 1996* c86.

288. Acrodin: absent (0); present (1).

*Patterson, 1982b* c12.

### Jaws, general
289. Jaws: absent (0); present (1).

*Dupret et al., 2014* c254.

290. Cosmine-like tissue in oral cavity: absent (0); present (1).

*Friedman, 2007a* c56.

### Dermal lower jaw bones
291. Dermal jaw plates on biting surface of jaw cartilages: absent (0); present (1).

*Brazeau, 2009* c48.

292. Dentary: absent (0); present (1).

New.

293. Large ventromesially directed flange of symphysial region of mandible: absent (0); present (1).

*Friedman, 2007b* c156.

294. Flange like extension of mandible composed of prearticular and Meckelian ossification: absent (0); present (1).

*Friedman, 2007b* c159.

295. Strong ascending flexion of symphysial region of mandible: absent (0); present (1).

*Friedman, 2007b* c155.

296. Labial pit: absent (0); present (1).

*Cloutier and Ahlberg, 1996* c80.

297. Infradentary: absent (0); present (1).

*Zhu et al., 2013* c204.

298. Number of infradentaries: one (0); two (1); more than 2 (2).
*Friedman, 2007b* c54; *King et al., 2017* c381.

299. Extent of infradentaries: along much of ventral margin of dentary (0); restricted to posterior half of dentary (1).
*Giles et al., 2015c* c93.

300. Infradentary foramina: present (0); absent (1).
*Ahlberg and Johanson, 1998b* 15; *King et al., 2017* 350.

301. Anterior end of prearticular: far from jaw symphysis (0); near jaw symphysis (1).
*Zhu and Schultze, 2001* c93.

302. Prearticular - dentary contact: present (0); absent (1).
*Cloutier and Ahlberg, 1996*, c96.

303. Principal coronoid: absent (0); present (1).
*Cloutier, 1996* c95.

304. Coronoids: present (0); absent (1).
*Schultze and Cumbaa, 2001* c46.

305. Number of coronoids: more than three (0); three (1).
*Ahlberg and Clack, 1998a* c4, *Zhu et al., 2009* c93.

306. Meckelian bone exposed immediately anterior to first coronoid: yes (0); no (1).
*Ahlberg and Clack, 1998a* c22.

307. Submandibulars: absent (0); present (1).
*Cloutier and Ahlberg, 1996* c104.

Palatoquadrate
308. Large otic process of the palatoquadrate: absent (0); present (1).
*Brazeau, 2009* c49.

309. Insertion area for jaw adductor muscles on palatoquadrate: ventral (0); lateral (1).
*Brazeau, 2009* c50.

310. Oblique ridge or groove along medial face of palatoquadrate: absent (0); present (1).
*Brazeau, 2009* c52.

311. Perforate or fenestrate anterodorsal (metapterygoid) portion of palatoquadrate: absent (0); present (1).
*Brazeau, 2009* c54.

312. Processus ascendens of palatoquadrate: absent (0); present (1).

*King et al., 2017* c389.

313. Fenestration of palatoquadrate at basipterygoid articulation: absent (0); present (1).

*Brazeau, 2009* c53.

314. Metapterygoid with developed medial ventral protrusion: absent (0); present (1).

*Zhu et al., 2013* c216.

315. Palatoquadrate fused with neurocranium: absent (0); present (1).

*Giles et al., 2015a* c101.

316. Jugular vein passes through cranioquadrate passage: absent (0); present (1).

*King et al., 2017* c126.

317. Autopalatine and quadrate: comineralized (0); separate mineralizations (1).

*Miles and Dennis, 1979* c22, *Giles et al., 2015b* c97.

318. Hyosuspensory eminence on quadrate: absent (0); present (1).

*Friedman, 2007a*, c55.

319. Dermal plates on mesial (lingual) surfaces of Meckels cartilage and palatoquadrate: absent (0); present (1).

*Zhu et al., 2013* c213.

320. Contact between palatoquadrate and dermal cheek bones: continuous contact of metapterygoid and autopalatine (0); metapterygoid and autopalatine contacts separated by gap between commissural lamina of palatoquadrate and cheek bones (1).

*Zhu et al., 2013* c215.

321. Number of fang pairs on ectopterygoid: one (0); two (1); none (2).

*Lu et al., 2012* c103.

322. Proportions of entopterygoid: anterior end level with processus ascendens (0); anterior end considerably anterior to processus ascendens (1).

*Lu et al., 2012* c104.

## Meckel's cartilage etc

323. Bilateral series of labial cartilages: absent (0); present (1).

*King et al., 2017* c393.

324. Pronounced dorsal process on Meckelian bone or cartilage: absent (0); present (1).

*Hanke and Wilson, 2004* c11; *Brazeau, 2009* c55.

325. Adductor fossa: open (0); reduced to narrow slot (1).

*Schultze, 2001* c69; *Ahlberg et al., 2006* c41.

326. Preglenoid process: absent (0); present (1).

*Davis et al., 2012* c52.

327. Biconcave glenoid on lower jaw: absent (0); present (1).

*Friedman and Brazeau, 2010* c17; *Zhu et al., 2013* c214.

328. Jaw articulation located on rearmost extremity of mandible: absent (0); present (1).

*Davis et al., 2012* c53.

329. Retroarticular process: absent (0); present (1).

*Lu et al., 2012* c163.

330. Symplectic articulation: absent (0); present (1).

*Friedman, 2007b* c160.

Scapulocoracoid

331. Scapular process of shoulder endoskeleton: absent (0); present (1).

*Brazeau, 2009* c105.

332. Ventral margin of separate scapular ossification: horizontal (0); deeply angled (1).

*Brazeau, 2009* c107.

333. Cross sectional shape of scapular process: flattened or strongly ovate (0); subcircular (1).

*Brazeau, 2009* c108.

334. Flange on trailing edge of scapulocoracoid: absent (0); present (1).

*Brazeau, 2009* c109.

335. Scapular process with posterodorsal angle: absent (0); present (1).

*Davis et al., 2012* c114.

336. Endoskeletal postbranchial lamina on scapular process: present (0); absent (1).

*Brazeau, 2009* c110.

337. Mineralization of internal surface of scapular blade: mineralised all around (0); unmineralised on internal face forming a hemicylindrical cross-section (1); unmineralised on lateral face forming a hemicylindrical cross-section (2).

*Brazeau, 2009* c111. According to the *Burrow and Rudkin, 2014* description, Nerepisacanthus has a scapulocroacoid mineralised only on the medial face. This necessitates a new character state (2).

338. Coracoid process: absent (0); present (1).

*Brazeau, 2009* c112.

339. Procoracoid mineralization: absent (0); present (1).

*Brazeau, 2009* c114.

340. Fin base articulation on scapulocoracoid: stenobasal (0); eurybasal (1).

*Brazeau, 2009* c113.

341. Perforate propterygium: absent (0); present (1).

*Patterson, 1982b* c15.

342. Endoskeletal supports in pectoral fin: multiple elements articulating with girdle (0); single element ('humerus') articulating with girdle (1).

*Zhu and Schultze, 2001* c175.

343. Triradiate scapulocoracoid: absent (0); present (1).

*Zhu and Schultze, 2001* c171.

344. Subscapular foramen: absent (0); present (1).

*Zhu and Schultze, 2001* c173.

345. Pectoral propterygium: absent (0); present (1).

*Zhu and Schultze, 2001* c176.

346. Horizontal plate of scapulocoracoid: absent (0); present (1).

*Patterson, 1982b* c17; *Friedman and Blom, 2006* c40.

### Dermal shoulder girdle
347. Macromeric dermal shoulder girdle: present (0); absent (1).

*Brazeau, 2009* c99.

348. Dermal shoulder girdle forming a complete ring around the trunk: present (0); absent (1).

*Brazeau, 2009* c101.

349. Median dorsal plate: absent (0); present (1).

*Brazeau, 2009* c103.

350. Number of MD plates: one (0); two (1); three (2).

*Trinajstic and Long, 2009* c445; *King et al., 2017* c445.

351. Pronounced internal crista (keel) on median dorsal surface of shoulder girdle: absent (0); present (1).

*Brazeau, 2009* c104.

352. Posteriorly spine on MD plate: absent (0); present (1).

*Carr and Hlavin, 2010* c37.

353. Anocleithrum: element developed as postcleithrum (0); element developed as anocleithrum sensu stricto (1).

*Gardiner and Schaeffer, 1989* cB2.

354. Anocleithrum sensu stricto: exposed (0); subdermal (1).

*Cloutier and Ahlberg, 1996* c112.

355. Pectoral fenestra completely encircled by dermal shoulder armour: present (0); absent (1).

*Brazeau, 2009* c102.

356. Dermal shoulder girdle composition: ventral and dorsal (scapular) components (0); ventral components only (1).

*Brazeau, 2009* c100.

357. Dorsal cleithrum (AL of the Placodermi), ventral cleithrum (AVL of the Placodermi) and pectoral spine (SP of the Placodermi): not fused (0); fused (1).

*Cloutier and Ahlberg, 1996* c161.

358. Relationship of clavicle to cleithrum: ascending process of clavicle overlapping cleithrum laterally (0); ascending process of clavicle wrapping round anterior edge of cleithrum, overlapping it both laterally and mesially (1).

*Cloutier and Ahlberg, 1996* c116.

359. Shape of dorsal blade of dermal shoulder girdle (either cleithrum or anterolateral plate): spatulate (0); pointed (1).

*Cloutier and Ahlberg, 1996* c115.

360. Chang"s apparatus: absent (0); present (1).

*King et al., 2017* c444.

361. Clavicles/interolateral plates: large plates, comparable in size to cleithrum (0); reduced to small semilunar plates, paired (1); unpaired semilunar plates (2).

*Jia et al., 2010* c44; *King et al., 2017* c443.

362. Median ventral trunk plate(s): absent (0); present (1).

*King et al., 2017* c447.

363. Extracleithrum: absent (0); present (1).

*Forey, 1998* c88.

364. Posterior dorsolateral (PDL) plate or equivalent: absent (0); present (1).

*Giles et al., 2015a* c187.

365. PL and PDL overlap: simple (0); insertion (1).

*Carr and Hlavin, 2010* c42.

366. Left and right PDL contact below MD: absent (0); present (1).

*King et al., 2017* c438.

367. PDL plate visible externally: yes (0); no (1).

*King et al., 2017* c439.

### Dermal pectoral fin

368. Scapular infundibulum: absent (0); present (1).

*Giles et al., 2015b* c190.

369. Pectoral fin base has large, hemispherical dermal component: absent (0); present (1).

*Brazeau, 2009* c121.

370. Pectoral fins covered in macromeric dermal armour: absent (0); present (1).

*Brazeau, 2009* c120.

371. Joint in macromeric armoured pectoral fin: unjointed (0); jointed (1).

*Jia et al., 2010* c27; *King et al., 2017* c441.

372 . Cd1 and Cd2 plates: in contact (0); separated (1).

*Jia et al., 2010* c28.

### Pectoral fin endoskeleton

373. Number of basals in polybasal pectoral fins: three or more (0); two (1).

*Giles et al., 2015b* c202.

374. Number of mesomeres in metapterygial axis: five or fewer (0); seven or more (1).

*Cloutier and Ahlberg, 1996* 123.

375. Biserial pectoral fin endoskeleton: absent (0); present (1).

*Giles et al., 2015b* c205.

376. Filamentous extension of pectoral fin from axillary region: absent (0); present (1).

*Giles et al., 2015b* c207.

377. Entepicondyle on humerus: present (0); absent (1).
*King et al., 2017* c418.

378. Distal articulation of propterygium: with fin rays (0); with a second enlarged plate (1); no articulation (2).
*King et al., 2017* c420.

### Pelvic fins and claspers
379. Pelvic fins: absent (0); present (1).
*Brazeau, 2009* c117.

380. Pelvic girdle with substantial dermal component: yes (0); no (1).
*Zhu et al., 2013* c252.

381. Claspers with large dermal J-shaped element: absent (0); present (1).
*Long et al., 2015* c258.

382. Claspers: absent (0); present (1).
*Brazeau, 2009* c119. Removed 'pelvic' from the definition, as placoderm claspers may not be associated with the pelvic fins (*Trinajstic et al., 2015*). However, for the purposes of this phylogenetic analysis, the claspers of placoderms and chondrichthyans can be considered primary homologues, as opposed to the coding in *Long et al., 2015* and *King et al., 2017*.

### Fin spines
383. Paired fin spines: absent (0); present (1).
*Brazeau, 2009* c125.

384. Pectoral fin spine small (bivalve-like): no (0); yes (1).
*King et al., 2017* c449.

385. Pelvic fin spine: absent (0); present (1).
*Zhu et al., 2013* c253.

386. Dorsal fin spines: absent (0); present (1).
*Brazeau, 2009* c123.

387. Anal fin spine: absent (0); present (1).
*Brazeau, 2009* c124.

388. Median fin spine insertion: shallow, not greatly deeper than dermal bones/scales (0); deep (1).
*Brazeau, 2009* c126.

389. Intermediate fin spines: absent (0); present (1).
*Brazeau, 2009* c127.

390. Intermediate spines when present: one pair (0); multiple pairs (1).
*Giles et al., 2015c* c219.

391. Intermediate spines with finlets: absent (0); present (1).
*King et al., 2017* c481.

392. Prepectoral fin spines: absent (0); present (1).
*Brazeau, 2009* c128.

393. Prepectoral spines form 'necklace': no (0); yes (1).
*King et al., 2017* c483.

394. Median ventral prepectoral spine: absent (0); present (1).
*King et al., 2017* c482.

395. Fin spine cross-section: Round or horseshoe shaped (0); Flat-sided, with rectangular profile (1).
*Giles et al., 2015c* c218.

396. Fin spines with ridges: absent (0); present (1).
*Brazeau, 2009* c129.

397. Fin spines with nodes: absent (0); present (1).
*Brazeau, 2009* c130.

398. Fin spines with rows of large retrorse denticles: absent (0); present (1).
*Davis et al., 2012* c134.

399. Expanded spine rib on leading edge of spine: absent (0); present (1).
*Giles et al., 2015c* c224.

400. Spine ridges: converging at the distal apex of the spine (0); converging on leading edge of spine (1).
*Giles et al., 2015c* c225.

### Median fins
401. Number of dorsal fins, if present: one (0); two (1).
*Coates and Sequeira, 2001* c10.

402. Posterior dorsal fin shape: base approximately as broad as tall, not broader than all of other median fins (0); base much longer than the height of the fin, substantially longer than any of the other dorsal fins (1).
*Giles et al., 2015a* c229.

403. Basal plate in dorsal fin: absent (0); present (1).
*Giles et al., 2015b* c230.

404. Branching radial structure articulating with dorsal fin basal plate: absent (0); present (1).
*Giles et al., 2015c* c231.

405. Anal fin: absent (0); present (1).
*Brazeau, 2009* c134.

406. Basal plate in anal fin: absent (0); present (1).
*Giles et al., 2015c* c233.

407. Spine-brush complex: absent (0); present (1).
*King et al., 2017* c479.

## Caudal fin
408. Horizontal caudal lobe: absent (0); present (1).
*Sansom, 2009* c70.

409. Triphycercal tail: absent (0); present (1).
*King et al., 2017* c452.

410. Caudal radials: extend beyond level of body wall and deep into hypochordal lobe (0); restricted to axial lobe (1).
*Davis et al., 2012* c138.

411. Supraneurals in axial lobe of caudal fin: absent (0); present (1).
*Giles et al., 2015a* c235.

412. Fringing fulcra: absent (0); present (1).
*Friedman, 2007b* c188.

413. Epichordal lepidotrichia in caudal fin: absent (0); present (1).
*Cloutier and Ahlberg, 1996* c134.

## Axial skeleton
414. Synarcual: absent (0); present (1).
*Brazeau, 2009* c132; *Davis et al., 2012* c135.

415. Series of thoracic supraneurals: absent (0); present (1).
*Cloutier and Ahlberg, 1996* c137.

## Fin webs
416. Longitudinal scale alignment in fin webs: present (0); absent (1).
*Giles et al., 2015a* c13.

417. Differentiated lepidotrichia: absent (0); present (1).
*Giles et al., 2013* c14.

418. Interlocking lepidotrichial segments: absent (0); present (1).
*Friedman, 2007b* c187.

## Postcranial plates

419. Scute-like ridge scales (basal fulcra): absent (0); present (1).

*Patterson, 1982b* c19.

420. Longitudinal rows of enlarged keeled scutes: absent (0); present (1).

*King et al., 2017* c484.

421. Series of median hexagonal scutes anterior to first dorsal fin: absent (0); present (1).

*King et al., 2017* c480.

## Scales

422. Body scale growth pattern: monodontode (0); polyodontode (1).

*Brazeau, 2009* c8.

423. Body scale growth concentric: absent (0); present (1).

*Brazeau, 2009* c9.

424. Postcranial scales with areal or appositional growth crowns: absent (0); present (1).

*Burrow et al., 2016* c260.

425. Body scales with peg-and-socket articulation: absent (0); present (1).

*Coates, 1999* c3.

426. Peg on rhomboid scale: narrow (0); broad (1).

*Patterson, 1982b* c5.

427. Anterodorsal process on scale: absent (0); present (1).

*Patterson, 1982b* c4.

428. Body scale profile: distinct crown and base demarcated by a constriction (neck) (0); flattened (1).

*Brazeau, 2009* c11.

429. Profile of scales with constriction between crown and base: neck similar in width to crown (0); neck greatly constricted, resulting in anvil-like shape (1).

*Giles et al., 2015c* c22.

430. Body scales with bulging base: absent (0); present (1).

*Brazeau, 2009* c12.

431. Body scales with flattened base: present (0); absent (1).

*Brazeau, 2009* c13.

432. Scales: macromeric (0); micromeric (1).

*Friedman and Blom, 2006* c34.

433. Flank scales alignment: vertical rows (0); oblique rows or hexagonal/ rhombic packing (1); disorganised (2).

*Davis et al., 2012* c14.

434. Basal pore in scales: absent (0); present (1).

*Giles et al., 2015c* c25.

435. Scales with well developed pores on ganoine surface: absent (0); present (1).

*Friedman and Blom, 2006* c35.

## Sensory lines, general

436. Sensory line network: preserved as open grooves (0); pass through canals enclosed within dermal bones (1).

*Brazeau, 2009* c40.

437. Sensory canals/grooves: contained within the thickness of dermal bones (0); contained in prominent ridges on visceral surface of bone (1).

*Giles et al., 2015c* c31.

## Sensory lines, snout

438. Course of ethmoid commissure: middle portion through median rostral (0); sutural course (1); through bone center of premaxillary (2).

*Zhu and Schultze, 2001* c43.

439. Ethmoid commissure fused into midline canal: absent (0); present (1).

*King et al., 2017* c320.

440. Infraorbital canal follows premaxillary suture: no (0); yes (1).

*Cloutier and Ahlberg, 1996* c100.

441. Anterior supraorbital canal: absent (0); present (1).

*Zhu and Gai, 2007* c38; *King et al., 2017* c309.

442. Semicircular pit-line: absent (0); present (1).

*Jia et al., 2010* c23.

443. Supraorbital sensory canal: absent (0); present (1).

*Zhu and Gai, 2007* c39; *King et al., 2017* c307.

444. Course of supraorbital canal: between anterior and posterior nostrils (0); anterior to both nostrils (1).

*Cloutier and Ahlberg, 1996* c98.

445. Contact of supraorbital and infraorbital canals: in contact rostrally (0); not in contact rostrally (1).

*Zhu et al., 2001* c34.

446. Contact between otic and supraorbital canals: not in contact (0); in contact (1).

*Cloutier and Ahlberg, 1996* c102.

447. Branching sensory canal system at posterior of supraorbital canals: absent (0); present (1).

*King et al., 2017* c306.

448. Posterior end of supraorbital canal: close to posterior and middle pit-lines (0); anterior to posterior and middle pit-lines (1); extends posterior to middle and posterior pit-lines (2).

*King et al., 2017* c292.

449. Anterior pit-line of dermal skull roof: absent (0); present (1).

*Giles et al., 2015a* c34.

450. Position of anterior pit-line: on paired median skull roofing bones over the otico-occipital division of braincase (0); on paired median skull roofing bones over the sphenoid division of braincase (1).

*Cloutier and Ahlberg, 1996* c103.

451. Course of supraorbital canal: straight (0); lyre-shaped (1).

*Zhu et al., 2013* c184.

452. Median dorsal canal: absent (0); present (1).

*Zhu and Gai, 2007* c43; *King et al., 2017* c310.

### Sensory lines, skull roof posterior

453. Infra-orbital sensory line: crosses lateral field (0); does not cross lateral field (1).

*Sansom, 2009* c31.

454. Festooned pattern of sensory canals: absent (0); present (1).

*Zhu and Gai, 2007* c2.

455. Branching of lateral transverse canal: absent (0); present (1).

*Zhu and Gai, 2007* c39.

456. Supraorbital canals and posterior pitlines cross as an X: no (0); yes (1).

*Long et al., 2015* c256; *King et al., 2017* c300.

457. Otic canal: runs through skull roof (0); follows edge of skull roof (1).

*Ahlberg and Johanson, 1998b* c66.

458. Extension of otic canal beyond infraorbital canal ('P' canal): absent (0); present (1).

*King et al., 2017* c319.

459. Otic canal extends through postparietals: absent (0); present (1).
*Cloutier and Ahlberg, 1996* c101.

460. Otic canal runs along mesial margin of marginal plate: absent (0); present (1).
*Dupret et al., 2009* c16.

461. Central sensory lines: absent (0); present (1).
*Dupret et al., 2009* c31.

462. Junction of posterior pitline and main lateral line: far in front of posterior margin of skull roof (0); close to posterior margin of skull roof (1).
*Zhu et al., 2013* c166.

463. Middle and posterior pit-lines on postparietal: posteriorly situated (0); mesially situated (1).
*Zhu et al., 2001* c39.

464. Position of middle and posterior pit-lines: close to midline (0); near the central portion of each postparietal (1).
*Zhu et al., 2006* c41.

465. Posterior pitline and postmarginal canal: staggered (0); confluent (1).
*King et al., 2017* c321.

466. Postmarginal canal: absent (0); present (1).
*King et al., 2017* c315.

467. Sensory line commissure across extrascapular bones: absent (0); present (1).
*King et al., 2017* c323.

### Sensory lines, cheek
468. Preopercular canal: absent (0); present (1).
*King et al., 2017* c316.

469. Preopercular canal meets main canal: no (0); yes (1).
*King et al., 2017* c317.

470. Horizontal sensory canal: absent (0); present (1).
*King et al., 2017* c314.

471. Jugal portion of infraorbital canal joins supramaxillary canal: present (0); absent (1).
*Davis et al., 2012* c17.

472. Supraoral canal: absent (0); present (1).
*King et al., 2017* c318.

473. Sensory canal or pit-line associated with maxilla: absent (0); present (1).

*Friedman, 2007b* c152.

## Sensory lines, jaw

474. Course of mandibular canal: not passing through most posterior infradentary (0); passing through most posterior infradentary (1).

*Cloutier and Ahlberg, 1996* c111.

475. Course of mandibular canal: passing through dentary (0); not passing through dentary (1).

*Patterson, 1982a* c7.

## Sensory lines, postcranial

476. Dorsal branch of main lateral line canal on PDL: absent (0); present (1).

*King et al., 2017* c325.

477. Sharp downward bend on PDL plate sensory line: absent (0); present (1).

*King et al., 2017* c326.

478. Sensory line canal: passes between or beneath scales (0); passes over scales and/or is partially enclosed or surrounded by scales (1); perforates and passes through scales (2).

*Friedman and Brazeau, 2010* c26; *Davis et al., 2012* c15.

## Visceral skeleton

479. Foramen in hyomandibular: absent (0); present (1).

*Friedman, 2007b* c163.

480. Endoskeletal hyoid rays: absent (0); present (1).

*King et al., 2017* c150.

481. Anteriormost unpaired element of branchial skeleton contacted by: one branchial arch only (0); two or more branchial arches (1).

*Dearden et al., 2019* c73.

482. Multiple unpaired branchial mineralizations: absent (0); present (1).

*Dearden et al., 2019* c74.

483. Interhyal: absent (0); present (1).

*Davis et al., 2012* c38.

484. Hypohyal: absent (0); present (1).

*Giles et al., 2015c* c75.

485. Endoskeletal urohyal: absent (0); present (1).

*Friedman, 2007b* c164.

486. Gill skeleton extends posteriorly beyond occiput: absent (0); present (1).

*Dearden et al., 2019* c67.

487. Sublingual rod: absent (0); present (1).

*King et al., 2017* c149.

488. Disposition of the interbranchial ridges of the oralobranchial chamber roof: oligobranchiate (0); orthobranchiate (1); nectaspidiform (2).

*Sansom, 2009* c62.

489. Number of branchial fossae: 5–7 (0); 9–17 (1); more than 20 (2).

*Zhu and Gai, 2007* c49.

## Homology-variable characters

The following characters have variable homology and the data matrix presents two options for their codings in placoderms. Both the marginal jaw bones (characters 490–498) and the palatal bones (characters 599–507) have the same set of dependent characters. Characters regarding the external, facial part of the maxilla and premaxilla are dependent on the presence of a facial lamina, which is absent in placoderms. These characters therefore are coded as inapplicable in placoderms regardless of homology, and do not need to be included in the homology-variable characters.

490. Premaxilla: absent (0); present (1).

New.

491. Maxilla: absent (0); present (1).

*Cloutier and Ahlberg, 1996* c19.

492. Facial laminae on (pre)maxilla: absent (0); present (1).

New.

493. Palatal laminae on (pre)maxilla: absent (0); present (1).

New.

494. Premaxilla fused in midline: absent (0); present (1).

*King et al., 2017* c372.

495. Pipe-like ridges on (pre)maxilla: absent (0); present (1).

*King et al., 2017* c373.

496. Premaxilla posterior process: absent (0); present (1).

New.

497. Fangs on premaxilla palatal lamina: absent (0); present (1).

New.

498. Maxilla fragmented into multiple bones: absent (0); present (1).
New.

499. Vomers: absent (0); present (1).
New.

500. Dermopalatine: absent (0); present (1).
New.

501. Facial laminae on vomer/dermopalatines: absent (0); present (1).
New.

502. Palatal laminae on vomer/dermopalatines: absent (0); present (1).
New

503. Vomers fused in midline: absent (0); present (1).
*King et al., 2017* c372.

504. Pipe-like ridges on vomer/dermopalatines: absent (0); present (1).
*King et al., 2017* c373.

505. Vomer posterior process: absent (0); present (1).
*Lu et al., 2012* c89; *Carr and Hlavin, 2010* c68; *King et al., 2017* c375.

506. Fangs on vomer palatal lamina: absent (0); present (1).
Adapted from *Ahlberg and Johanson, 1998b* c24.

507. Dermopalatine fragmented into multiple bones: absent (0); present (1).
*Lu et al., 2012* c106; *King et al., 2017* c378.

