## [Decision Letter]

**Acceptance summary:**

The phylogeny of early jawed vertebrates has been discussed in depth during recent years, along with many new fossils emerging and the re-examination of the published materials using novel techniques. The key to the issue is how to interpret the morphological characters of the studied taxa, especially when large morphological gaps exist, which impedes the use of the topological correspondence criterium. This manuscript presents a conceptually interesting approach to a type of intractable homology problem that we encounter from time to time in morphological phylogenetics, and will engender a lively and healthy debate in early vertebrate phylogeny.

**Decision letter after peer review:**

Thank you for submitting your article "A Bayesian approach to dynamic homology of morphological characters and the ancestral phenotype of jawed vertebrates" for consideration by *eLife*. Your article has been reviewed by two peer reviewers, including Min Zhu as the Reviewing Editor and Reviewer #1, and the evaluation has been overseen by George Perry as the Senior Editor. The following individual involved in review of your submission has agreed to reveal their identity: Per E Ahlberg (Reviewer #2).

The reviewers have discussed the reviews with one another and the Reviewing Editor has drafted this decision to help you prepare a revised submission.

Summary:

The phylogeny of early jawed vertebrates has been discussed in depth during recent years, along with many new fossils emerging and the re-examination of the published materials using novel techniques. The key to the issue is how to interpret the morphological characters of the studied taxa, especially when large morphological gaps exist, which impedes the use of the topological correspondence criterium. Another puzzling problem is the selection between the diverged optimal phylogenetic hypotheses resulted from different methods (parsimony vs. bayesian / Bayesian morphological clock methods). In King et al., 2017, the same character matrix yielded different phylogenetic scenarios: (1) "parsimony analysis showed that placoderm paraphyly and monophyly are essentially equally parsimonious", (2) "the Bayesian morphological clock methods resurrect placoderm monophyly". This manuscript follows this thinking thread and proposes a Bayesian approach to dynamic homology of morphological characters using a matrix modified from that in King et al., 2017. Their discussions on the ancestral phenotype of jawed vertebrate are impressive. Although the dynamic homology approach has raised many controversies, it has been used (if not widely) in some phylogenetic studies. This manuscript provides a new case to use the dynamic homology approach and takes a highly innovative approach to a particularly vexing homology problem in early vertebrate phylogeny and evolution. In this regard, this manuscript has its value to be considered by *eLife* and will engender a lively and healthy debate.

Essential revisions:

However, the manuscript still needs revisions to respond to the following comments from the two reviewers.

1) The character matrix is still the core of the phylogenetic analysis. The matrix herein was revised from that of King et al., 2017. Understandably, the new matrix was updated with the addition of several new taxa. Meanwhile, four taxa were removed with no interpretation. If one or all of these four taxa were not removed from the matrix, what are the phylogenetic results? Considering the weak phylogenetic supports for several key nodes, some small changes of taxon inclusion and character coding might much impact on the phylogenetic results.

2) With the new matrix, what is the most parsimonious tree from the parsimony analysis? Is it same as or different from the optimal tree from the tip-dated homoplasy partitioned analysis?

3) The specific problem that the authors address, the homology of the dentigerous jaw bones of placoderms, is actually part of a broader problem area that could probably be tackled in a similar way in future. Traditionally, the entire macromeric dermal skeleton of placoderms has been regarded as non-homologous with that of osteichthyans, and has been given its own terminology ("central", "submarginal" and so forth), but many of us now regard the placoderm and osteichthyan skeletons as homologous. For example, the central appears to correspond to the osteichthyan postparietal and the submarginal to the opercular. It might be nice to mention this broader problem briefly in the Introduction.

4) It would be good to mention Chen et al., 2017, in the Introduction and Discussion, because it shows that the inner dental arcade of the stem osteichthyan *Lophosteus* consists of numerous small cushion-shaped elements, completely unlike arthrodire gnathal plates in appearance. This provides further support for the conclusion reached by the authors. The assignment of these bones to the inner dental arcade (not the gill arches) has been controversial in some quarters, but as you know, the (unpublished) articulated head of Megamastax confirms Chen et al.'s interpretation.

5) The reviewer is not a big fan of tip-dating, especially with a taxon sample like this that includes the earliest known articulated gnathostomes and is likely to be strongly affected by powerful but unknown taphonomic/environmental/biogeographic filters. It also seems to the reviewer that the specific taxa chosen for the analysis could have a big effect. For example, the earliest antiarchs included in the analysis are *Yunnanolepis* and *Parayunnanolepis*, both from the Lochkovian (Early Devonian), but the earliest known antiarch is *Shimenolepis* from the Ludlow. How would the analysis be affected by the inclusion of this taxon?

6) (Relating to the second paragraph of the Discussion). The biting surfaces of the jaw bones of *Entelognathus* are not tuberculated, they only look that way because of the mechanical preparation. A synchrotron scan of the holotype the reviewer made a while back shows that the dentition actually consists of needle-like teeth. This isn't published yet and cannot be cited, but the authors can avoid making an incorrect statement.

7) In the discussion about the fate of facial and oral laminae, it would be good to mention that the marginal jawbones of the stem osteichthyans *Andreolepis* (Botella et al., 2007; Chen et al., 2016) and *Lophosteus* (Botella et al., 2007; Chen et al., 2020) both have more strongly developed oral laminae than those of crown osteichthyans. This fits nicely with the conclusions of the authors.

8) Why does the analysis find strong support for anterior-ventral nasal capsules as the primitive condition for jawed vertebrates, when the two consecutive outgroups (osteostracans and galeaspids) both have posterior-dorsal nasal capsules?

---

## [Author Response]

Essential revisions:However, the manuscript still needs revisions to respond to the following comments from the two reviewers.1) The character matrix is still the core of the phylogenetic analysis. The matrix herein was revised from that of King et al., 2017. Understandably, the new matrix was updated with the addition of several new taxa. Meanwhile, four taxa were removed with no interpretation. If one or all of these four taxa were not removed from the matrix, what are the phylogenetic results? Considering the weak phylogenetic supports for several key nodes, some small changes of taxon inclusion and character coding might much impact on the phylogenetic results.

We have now added the complete justification for the removal of four taxa in Appendix 2. Note also that all taxa would be unknown for the jaw bone characters under discussion, and therefore their removal would not affect this. The similarity of the trees found to those in King et al., 2017, shows that the results are not greatly affected by taxon inclusions and removals. The text added to the Appendix is as follows:

“Wuttagoonaspis fletcheri

The description of the braincase of *Wuttagoonaspis* in Ritchie, 1973, does not match direct observations of specimens (by B. King). […] *Raynerius* was added as an alternative early acintopterygian with better quality preservation of many features.”

2) With the new matrix, what is the most parsimonious tree from the parsimony analysis? Is it same as or different from the optimal tree from the tip-dated homoplasy partitioned analysis?

Similar to the results of King et al., 2017, the new matrix when analysed under parsimony gives essentially similar tree lengths for placoderm paraphyly (1285) and monophyly (1286). As stated in the Introduction the purpose of this paper is to explore the hypothesis of Zhu et al., within the framework of placoderm monophyly as supported by tip dating. It is beyond the scope of this paper to explore the differences between methods, which was already covered at length in the King et al., 2017, paper, and there is nothing new to add here.

To clarify this, we have added the following paragraph to the Discussion:

“The results of our analysis are contingent on a phylogenetic hypothesis, in particular the monophyly of core placoderms, which is only strongly supported under a Bayesian tip-dating approach. […] The hypothesis of placoderm paraphyly (Brazeau, 2009; Davis et al., 2012; Zhu et al., 2013), implies a radically different scenario for character evolution (Dupret et al., 2014), in which the maxillate placoderms are not representative of ancestral conditions.”

3) The specific problem that the authors address, the homology of the dentigerous jaw bones of placoderms, is actually part of a broader problem area that could probably be tackled in a similar way in future. Traditionally, the entire macromeric dermal skeleton of placoderms has been regarded as non-homologous with that of osteichthyans, and has been given its own terminology ("central", "submarginal" and so forth), but many of us now regard the placoderm and osteichthyan skeletons as homologous. For example, the central appears to correspond to the osteichthyan postparietal and the submarginal to the opercular. It might be nice to mention this broader problem briefly in the Introduction.

Revised to:

“The discovery of maxillate placoderms reignited debates about the homology of placoderm and osteichthyan skull bones (Zhu et al., 2013; Zhu et al., 2016), and a new hypothesis regarding the homology of arthrodiran gnathal plates was proposed (Zhu et al., 2016; Zhu et al., 2019).”

We also added the following to the Discussion:

“In this manuscript we have applied the method to placoderm jaw bones, but it could also potentially be used to examine skull roof homologies in the future.”

4) It would be good to mention Chen et al., 2017, in the Introduction and Discussion, because it shows that the inner dental arcade of the stem osteichthyan Lophosteus consists of numerous small cushion-shaped elements, completely unlike arthrodire gnathal plates in appearance. This provides further support for the conclusion reached by the authors. The assignment of these bones to the inner dental arcade (not the gill arches) has been controversial in some quarters, but as you know, the (unpublished) articulated head of Megamastax confirms Chen et al.'s interpretation.

We have added the following to the Discussion:

“the inner dental arcade of the stem osteichthyan *Lophosteus* consists of many “tooth cushions” bearing no resemblance to arthrodire gnathal plates (Chen et al., 2017).”

5) The reviewer is not a big fan of tip-dating, especially with a taxon sample like this that includes the earliest known articulated gnathostomes and is likely to be strongly affected by powerful but unknown taphonomic/environmental/biogeographic filters. It also seems to the reviewer that the specific taxa chosen for the analysis could have a big effect. For example, the earliest antiarchs included in the analysis are Yunnanolepis and Parayunnanolepis, both from the Lochkovian (Early Devonian), but the earliest known antiarch is Shimenolepis from the Ludlow. How would the analysis be affected by the inclusion of this taxon?

To test the effect of a Silurian antiarch on the results, we reanalysed the data but with a Ludlow age assigned to *Yunnanolepis*. This is a conservative test since it is unlikely that if *Shimenolepis* were known in more detail that it would be identical in characters to *Yunnanolepis*. This analysis reveals no major difference in tree topology.

We have added the following section to Appendix 1:

“Sensitivity analysis

Bayesian tip-dated analysis may be sensitive to incomplete taxon sampling (O’Reilly and Donoghue, 2020). […] However, future studies should aim to further explore the effect of taxon sampling on results.”

6) (Relating to the second paragraph of the Discussion). The biting surfaces of the jaw bones of Entelognathus are not tuberculated, they only look that way because of the mechanical preparation. A synchrotron scan of the holotype the reviewer made a while back shows that the dentition actually consists of needle-like teeth. This isn't published yet and cannot be cited, but the authors can avoid making an incorrect statement.

Revised to:

“lacking large teeth on their jaw bones and in the case of *Entelognathus*, immovable eyes (Zhu et al., 2013).”

7) In the discussion about the fate of facial and oral laminae, it would be good to mention that the marginal jawbones of the stem osteichthyans Andreolepis (Botella et al., 2007; Chen et al., 2016) and Lophosteus (Botella et al., 2007; Chen et al., 2020) both have more strongly developed oral laminae than those of crown osteichthyans. This fits nicely with the conclusions of the authors.

Added the following:

“The stem osteichthyans *Lophosteus* and *Andreolepis* show a possibly intermediate condition, in which the marginal jaw bones have internal (oral or palatal) laminae that are more strongly developed compared to other osteichthyans (Botella et al., 2007; Chen et al., 2016; Chen et al., 2020; Cunningham et al., 2012).”

8) Why does the analysis find strong support for anterior-ventral nasal capsules as the primitive condition for jawed vertebrates, when the two consecutive outgroups (osteostracans and galeaspids) both have posterior-dorsal nasal capsules?

Revised to:

“Due to the nested position of acanthothoracids and antiarchs within a monophyletic core placoderms, we find strong support for anterior-ventral nasal capsules and lateral eyes in the gnathostome ancestor.”